# Sex-specific regulation of inhibition and network activity by local aromatase in the mouse hippocampus

Alicia Hernández-Vivanco [1,5], Nuria Cano-Adamuz [1,5], Alberto Sánchez-Aguilera [1,2], Alba González-Alonso [1], Alberto Rodríguez-Fernández [1], Íñigo Azcoitia [3,4], Liset Menendez de la Prida [1] & Pablo Méndez [1✉]

Cognitive function relies on a balanced interplay between excitatory and inhibitory neurons (INs), but the impact of estradiol on IN function is not fully understood. Here, we characterize the regulation of hippocampal INs by aromatase, the enzyme responsible for estradiol synthesis, using a combination of molecular, genetic, functional and behavioral tools. The results show that CA1 parvalbumin-expressing INs (PV-INs) contribute to brain estradiol synthesis. Brain aromatase regulates synaptic inhibition through a mechanism that involves modification of perineuronal nets enwrapping PV-INs. In the female brain, aromatase modulates PV-INs activity, the dynamics of network oscillations and hippocampal-dependent memory. Aromatase regulation of PV-INs and inhibitory synapses is determined by the gonads and independent of sex chromosomes. These results suggest PV-INs are mediators of estrogenic regulation of behaviorally-relevant activity.

[1] Instituto Cajal (CSIC), Av Dr. Arce 37, 28002 Madrid, Spain. [2] Department of Physiology, Faculty of Medicine, Universidad Complutense de Madrid IdISSC, Avda Complutense s/n, 28040 Madrid, Spain. [3] Department of Cell Biology, Universidad Complutense de Madrid, C José Antonio Nováis 12, 28040 Madrid, Spain. [4] Centro de Investigación Biomédica en Red de Fragilidad y Envejecimiento Saludable (CIBERFES), Instituto de Salud Carlos III, Madrid, Spain. [5] These authors contributed equally: Alicia Hernández-Vivanco, Nuria Cano-Adamuz. ✉email: pmendez@cajal.csic.es

Estradiol, a sex hormone involved in the control of female reproductive system, is a potent modulator of neuronal function[1,2]. The enzyme aromatase, expressed in the ovaries and neurons from different brain regions, catalyzes aromatization of testosterone to produce estradiol[3]. Aromatase-derived estradiol deeply impacts neuronal and synaptic activity, plasticity and cognitive function[4]. Indeed, preventing estradiol synthesis by pharmacological inhibition of aromatase, a widely used strategy for treating a diverse range of human diseases[5], reduces performance of laboratory animals[6,7] and humans[8–10] on different cognitive tasks. It is thus important to define the sources and targets of estradiol in the brain to understand basic mechanisms of cognitive processes and to identify potential therapeutic strategies based on aromatase inhibitors and selective estrogen receptor modulators[11].

The normal operation of the hippocampus, a brain region essential for learning and memory, relies on a balanced interplay between excitatory and inhibitory neurons (INs)[12]. Hippocampal CA1 excitatory pyramidal (PYR) neurons express aromatase[13], being a local source of estradiol that regulates gene expression and synaptic plasticity[14–16]. Yet, estrogenic effects on the hippocampal inhibitory system are relatively less explored[17]. Estrous cycle and pubertal-related changes in circulating estrogens regulate inhibition in basolateral amygdala and cortex[18–20], but the effect on CA1 synaptic inhibition is not fully defined. In brain slices and cell cultures, exogenously applied estradiol acts directly and rapidly to suppress inhibitory synapse activity in CA1 PYR neurons of female rat and mice[21–23], suggesting effects on hippocampal inhibition. However, the case of hippocampal INs as local estradiol sources and the role of brain estradiol synthesis in regulating hippocampal synaptic inhibition in vivo remain to be investigated.

A diverse group of INs expressing the calcium-binding protein parvalbumin (PV) provide a substantial fraction of synaptic inhibition to CA1 excitatory PYR neurons through the release of ɣ-aminobutyric acid (GABA)[24]. This heterogeneous group of INs comprises perisomatic-targeting basket cells and axoaxonic PV-INs as well as dendrite-targeting bistratified and Oriens-lacunosum moleculare (O-LM) INs[25]. The activity of PV-INs is determined by diverse array of ion channels[26–28]. In addition, perineuronal nets (PNNs), specialized extracellular proteoglycan aggregates, regulate the function of PV basket cells (BC)[29]. The activity of PV-INs varies according to physiological state (e.g. wakefulness, sleep[30]) and cognitive demands and is tightly coupled to specific aspects of behavior, such as locomotion[31–33]. In turn, the extreme divergence of PV-INs output connectivity allows them to coordinate the activity of large ensembles of neurons giving rise to different forms of oscillations, including theta and gamma oscillations and sharp-wave ripples (SWRs). Hippocampal oscillations underlie cognitive functions[34,35] and are basic mechanisms of information processing and memory consolidation[36]. Peripheral estrogens regulate hippocampal CA1 network oscillations associated to specific behavioral states[37,38], but the role of brain synthesized estradiol in the regulation of CA1 network activity has not been identified.

Here, we describe the expression and functional regulation of aromatase in a major type of hippocampal IN in mice. We found that CA1 PV-INs contribute to estradiol synthesis within the brain. In turn, brain aromatase influences CA1 synaptic inhibition through regulation of PNNs surrounding PV-INs. In vivo, aromatase has a direct impact on locomotion related PV-IN activity, hippocampal gamma and theta oscillations and SWRs dynamics, which strongly rely on PV-IN activity. Interestingly, aromatase regulation of CA1 synaptic inhibition and PV-INs is only present in female mice. We determined the contribution of sex chromosomes and gonads to sex-specific estradiol regulation of PV-INs. Our results suggest that gonadal, but not genetic sex, determine PV-INs as sources and targets of estradiol in the female brain.

## Results

**Aromatase mRNA, protein and its enzymatic product, 17-β-estradiol in female CA1 PV-INs.** While the expression of aromatase in CA1 pyramidal (PYR) neurons is well established[13,14,39], less is known about INs. To investigate the contribution of hippocampal INs to local estradiol synthesis, we studied the expression of aromatase mRNA and protein in female PV-expressing CA1 INs, a major class of hippocampal INs. We performed in situ hybridization in brain sections from adult female mice with specific probes against aromatase gene *cyp19a1* mRNA. We visualized PV-expressing neurons in CA1 by infecting PV-Cre mice with adeno-associated viruses (AAVs) expressing a green fluorescent protein (EYFP) in a Cre recombinase-dependent manner (Fig. 1A). Aromatase mRNA was present in identified PV-INs (PV, Fig. 1B). Signals from aromatase mRNA and PV-INs showed significantly higher colocalization levels compared with chance colocalization observed after relative rotation of the images (two-tailed paired *t*-test, $t(25) = 9.051$, $p < 0.0001$, $n = 26$ cells from 2 mice). Signal was not detected when the probe was omitted (Negative control, Fig. 1C). High levels of aromatase mRNA were found in the medial amygdala (Positive control, Fig. 1C) in agreement with previous reports[40]. Quantification of mRNA signal showed that aromatase mRNA signal in CA1 PV-INs was higher compared with *stratum pyramidale* (SP) but lower compared with medial amygdala (MeA, Fig. 1D). Simultaneous immunohistochemical localization of aromatase and PV protein in CA1 area of female mice showed aromatase expression in scattered neurons in *pyramidale* and *oriens strata*. Many of these cells were also positive for PV staining (Fig. 1E). We compared aromatase expression in PV-INs, CA1 PYR excitatory neurons of the *stratum pyramidale* and somatostatin (SST)-expressing neurons, another prominent subtype of CA1 INs. Although we observed aromatase expression in all cell types, the expression of aromatase protein was higher in PV-INs compared with PYR neurons (Fig. 1F) and SST-INs (Fig. S1A, B).

Finally, we assessed the presence of 17β-estradiol (βE2), the main enzymatic product of aromatase in the CA1 region using a specific antibody[41]. Co-staining with the IN marker PV showed βE2 presence in this IN subpopulation (Fig. 1G).

Altogether, these results show that aromatase mRNA, as well as the associated protein and enzymatic product are all present in hippocampal CA1 PV-INs, suggesting estradiol synthesis by this IN population.

**Aromatase expression in PV basket cells.** Molecular criteria allow the distinction of PV-INs subtypes with precise morphological and physiological characteristics[42,43]. For example, perineuronal nets (PNNs) surround almost all PV basket cells and, to a lesser or no extent, PV bistratified, PV axoaxonic cells and PV O-LM neurons[44,45]. On the other hand, the chromatin and gene expression regulatory protein SATB1 is expressed in CA1 PV basket cells and bistratified cells, but not in axoaxonic INs[45,46]. In order to clarify whether aromatase expression is subtype-specific, we determined the expression of aromatase together with PNNs or SAT1B, bona fide cell-type specific markers of PV-IN population.

To this purpose, we used *Wisteria floribunda* agglutinin (WFA) that detects N-acetyl-galactosamine residues in chondroitin sulfate chains as markers of PNNs in CA1 PV-INs. We observed that 83% of PV-INs show PNNs (PNN$^+$) with a variable range of

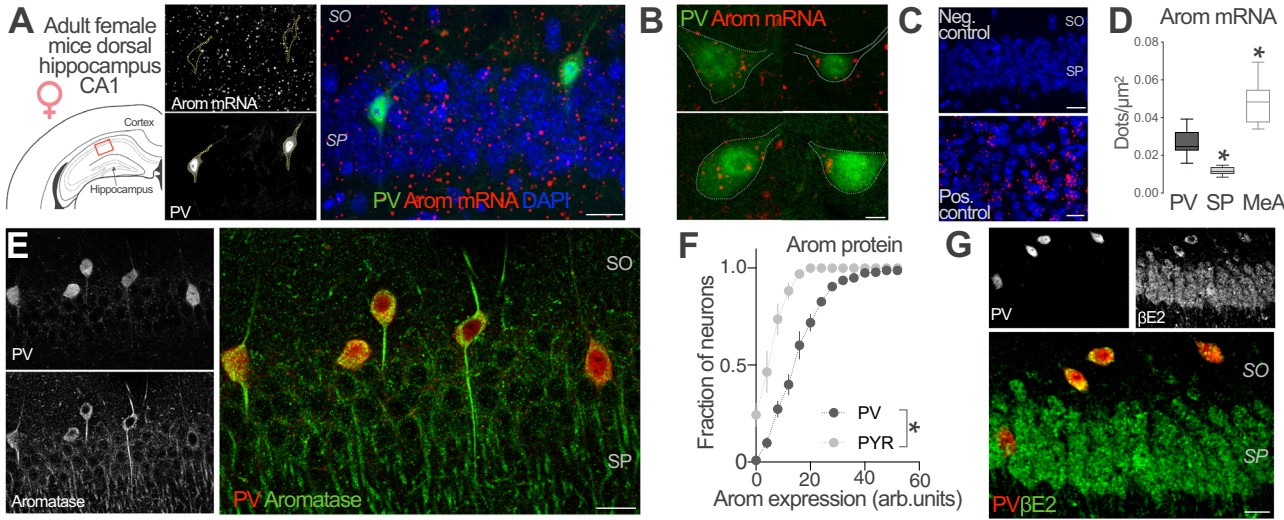

**Fig. 1 Aromatase expression and βE2 presence in PV-INs of female mice. A** Representative image of aromatase (arom) mRNA in situ hybridization in dorsal hippocampus CA1 region (approximate location, red square, left schema) of an adult parvalbumin-Cre (PV-Cre) female mice infected with AAVs carrying a Cre-dependent EYFP expression construct. EYFP labeled parvalbumin inhibitory neurons (PV-INs) and mRNA single-channel images are represented in gray scale in the middle part of the panel. SO stratum oriens, SP stratum pyramidale. Scale bar: 20 µm. **B** Aromatase mRNA signal (red dots) was detected in hippocampal PV EYFP+ neurons located in the SO and SP of the dorsal CA1 area. Scale bar: 5 µm. **C** No signal was detected in CA1 when probe was omitted (Neg. control). High aromatase mRNA signal (red dots) was detected in medial amygdala, a positive control (Pos. control) for aromatase expression. Scale bar: 20 µm. **D** Aromatase mRNA dot density analysis in CA1 PV-INs, CA1 stratum pyramidale (SP) and Medial Amygdala (MeA). Aromatase mRNA expression in PV-INs was higher compared with SP but lower than in MeA. One-way ANOVA, $F(2, 42) = 49.54$, $p < 0.0001$. Bonferroni's comparison tests, PV-SP $p < 0.0001$, PV-MeA $p < 0.0001$; $n = 26$, 9 and 10 PV-INs, SP and MeA neurons respectively, from 2 mice. Whiskers represent median and 10–90 percentiles. **E** Simultaneous immunohistochemical detection of aromatase protein (green) and PV (red) in CA1 region of female mice. Single-channel images are reproduced in gray scale in the left part of the panel. Scale bar: 20 µm. **F** Frequency distribution analysis of aromatase protein staining intensities in PV-INs (black) and pyramidal neurons (PYR, gray), arbitrary units (arb. units). Two-way ANOVA, $F(1, 4) = 74.05$, $p = 0.001$; $n = 3$ mice. Graph represents mean ± SEM. **G** Presence of 17-β-estradiol (βE2, green) in PV-IN (red) of the CA1 region of female mice was assessed using immunohistochemical analysis. Single-channel images are represented in gray scale in the upper part of the panel. Scale bar: 20 µm. *$p < 0.05$. Source data are provided as a Source Data file.

WFA staining intensities (Fig. 2A, B). Consistent with previous reports[45], all analyzed CA1 PNN+ cells expressed PV. Aromatase expression was higher in PNN+ PV-INs compared with PNN-lacking PNN− PV-INs (Fig. 2B, whisker plot). The proportion of aromatase expressing (Arom+) neurons was 79% in the population of PNN+ PV-INs (Fig. 2C, upper plot), while the remaining 21% were Arom−. In contrast, analysis of the population of PV-INs lacking PNNs (PNN− PV-INs) showed that 26% were Arom+ and 74% were Arom− (Fig. 2C, lower plot, 170 PV-INs from 3 female mice). On the other hand, SATB1 expression was found on 78% of female CA1 PV-INs (Fig. 2D, E). Aromatase expression was higher in SATB1+ PV-INs compared with SATB1− PV-INs (Fig. 2E, whisker plot). The proportion of Arom+ was 70% in the population of SATB1+ PV-INs (Fig. 2F, upper plot), the remaining 30% were Arom−. Analysis of PV-INs lacking SATB1 expression (SATB1− PV-INs) showed that 37% of SATB1− PV-INs were Arom+ and 63% were Arom− (Fig. 2F, lower plot, 245 PV-INs from 3 female mice).

These results show that aromatase expression differs between molecularly defined PV-INs subtypes with higher expression in PNN+ and SATB1+ PV basket cells.

### Brain aromatase regulates synaptic inhibition in CA1 pyramidal neurons of female mice.

Gonadal[47,48] and brain[14,15] estradiol synthesis regulates hippocampal excitatory activity but the impact on regulating inhibitory neuronal function remains less understood. We investigated the consequences of in vivo blockade of different estradiol sources (gonadal and extragonadal) on synaptic inhibition in CA1 PYR in female mice. We treated intact and ovariectomized (OVX) female mice with the aromatase blocker letrozole (0.5 mg/kg, one daily intraperitoneal injection during 5 days, Arom Block, Fig. 3A). Letrozole inhibits peripheral and central aromatase since it crosses the blood–brain barrier after systemic application[16]. A prolonged (5–7 days) aromatase blockade period is required to maximally block hippocampal synaptic plasticity (i.e. Long Term Potentiation) in intact and OVX female mice[49,50]. On the 5th day of treatment, we performed whole-cell patch-clamp recordings of spontaneous Inhibitory Post-Synaptic Currents (sIPSCs) from CA1 PYR neurons in acutely prepared brain slices.

Intact female mice with systemic aromatase blockade showed increased sIPSCs frequency in CA1 PYR neurons compared with vehicle treated intact mice (Intact, Arom Block, Fig. 3B, D). In contrast, no difference in sIPSCs amplitude was observed. Similarly, blockade of aromatase in OVX female mice significantly increased the frequency of sIPSCs and produce no evident changes in amplitude (OVX, Arom Block, Fig. 3C, D). Moreover, daily intracerebral administration of the aromatase blocker through a guide cannula implanted in the lateral ventricle during 5 days increased sIPSCs frequency but not amplitude in CA1 PYR neurons in female OVX mice (Fig. 3E, F). These results show that blockade of aromatase increases sIPSCs in CA1 PYR neurons and suggest that estradiol synthesis within the brain regulates synaptic inhibition in female mice.

To confirm the specificity of these results, we performed recovery experiments and determined the ability of exogenous estradiol to counteract the effect of aromatase blockade on sIPSCs recorded from female CA1 PYR neurons. To this purpose, we administered 17β-estradiol (βE2, 2 mg/kg, Recovery) or vehicle to

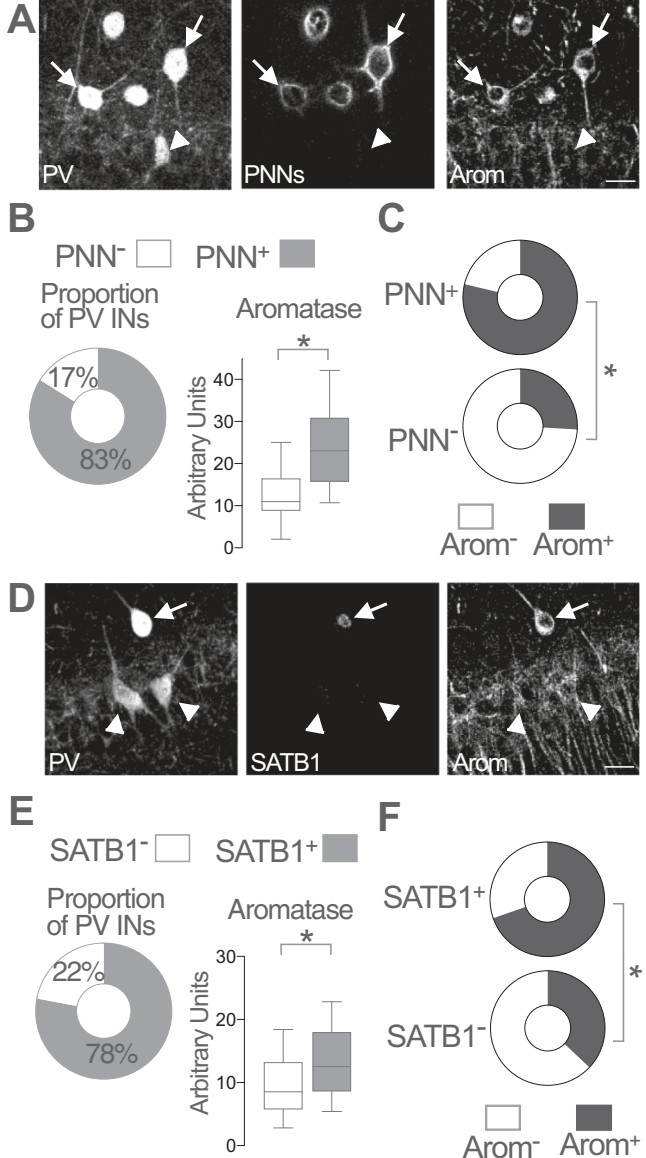

**Fig. 2 Aromatase expression in PNNs and SATB1-expressing PV-IN.**
**A** Representative images of parvalbumin (PV), aromatase (Arom) and WFA staining (perineuronal nets, PNNs) in adult female mice CA1 region. WFA staining was used to reveal the presence (arrows) or absence (arrowheads) of PNNs surrounding parvalbumin inhibitory neurons (PV-INs). Scale bar: 20 μm. **B** Proportion of PV-INs surrounded by PNNs (left plot). Higher aromatase levels were detected in PNN+ PV-INs as compared with PNN− PV-INs (right plot). Unpaired two-tailed t-test, *p < 0.0001, t(168) = 4.3; n = 170 PV-IN from 3 female mice. **C** Proportion of aromatase expressing (Arom+) neurons was higher in PNN+ PV-INs (upper plot) compared with PNN− PV-INs (lower plot). Two-sided Fisher's exact test, p < 0.0001; n = 170 PV-IN from 3 female mice. **D** Representative images of PV, aromatase and SATB1 staining in adult female mice CA1 region. Arrow points to a SATB1+ PV-IN. Arrowheads point to SATB1− PV-INs. Scale bar: 20 μm. **E** Proportion of PV INs expressing SATB1 (left plot). Higher aromatase levels were detected in SATB1+ PV-INs as compared with SATB1− PV-INs (right plot). Two-tailed Mann–Whitney test, U = 3304, p < 0.0001; n = 245 PV-INs from 3 female mice. **F** Proportion of aromatase expressing (Arom+) neurons was higher in SATB1+ PV-INs (upper plot) compared with SATB1− PV-INs (lower plot). Two-sided Fisher's exact test, p < 0.0001; n = 245 PV-IN from 3 female mice. Whisker in plots represent median and 10–90 percentiles. *p < 0.05. Source data are provided as a Source Data file.

intact female mice treated with aromatase blocker letrozole (Arom Block, Fig. 3G). βE2 treatment in female mice with pharmacological blockade of estradiol synthesis significantly reduced sIPSCs frequency but did not alter sIPSCs amplitude compared with vehicle-injected animals (Fig. 3H).

Altogether, these experiments show that estradiol synthesized by aromatase expressed in the brain reduces CA1 synaptic inhibition in female mice.

**PNNs are required for aromatase regulation of CA1 synaptic inhibition.** We then explored what aspects of the IN-PYR microcircuit may be involved in aromatase regulation of sIPSCs in female mice. Thus, we determined inhibitory synaptic activity in CA1 PYR neurons in the presence of the sodium channel inhibitor tetrodotoxin (TTX) to prevent action potentials, or in the presence of the wide spectrum glutamate receptor blocker kynurenic acid to block excitatory synaptic activity. In these two different conditions, we were unable to detect significant changes in frequency or amplitude of sIPSCs between control and aromatase blocked OVX female mice (Fig. S2). This suggest that aromatase control of female CA1 synaptic inhibition involves regulation of IN excitability.

PV-INs are a major class of hippocampal INs and form a large fraction of inhibitory synapses onto CA1 PYR neurons[51]. We focused on PNNs, which act as direct regulators of neuronal and synaptic excitability of this IN subtype[52–54]. We hypothesized that aromatase regulation of synaptic inhibition may involve modifications of these extra-cellular structures. To address this, we first evaluated the effect of pharmacological blockade of aromatase on PV-INs PNNs in female mice (Fig. 4A). Frequency distribution analysis of WFA staining intensities in intact female CA1 PV-INs showed that systemic aromatase blockade increased the intensity of WFA staining surrounding female PV-INs (Fig. 4B). Intracerebral administration of the aromatase blocker through a guide cannula implanted in the lateral ventricle similarly increased the intensity of WFA staining surrounding OVX female PV-INs (Fig. S3A, B). These results suggest that blockade of brain aromatase promotes the increase of PNNs around CA1 PV-INs in female mice.

We next investigated the involvement of PNNs in aromatase control of female synaptic inhibition. For this, we performed intrahippocampal injections of chondroitinase ABC (500 nl of ChABC 40 units/ml solution), an enzyme that degrades chondroitin sulfate chains present in PNNs. ChABC treatment in OVX female mice produced a decrease in WFA staining in CA1 PV-INs, evident 3 days after treatment (Fig. S3C–E). We evaluated the effect of aromatase blockade on sIPSCs frequency recorded in CA1 PYR neurons in OVX mice with intrahippocampal injections of chondroitinase ABC or vehicle (Fig. 4C). While ChABC treatment in OVX female mice did not produce significant sIPSCs frequency changes when compared with control mice, it completely prevented the effect of aromatase blockade on sIPSCs frequency (Fig. 4D). We observed no significant differences in the amplitude of sIPSCs between the experimental groups (Fig. 4D).

Altogether, our results suggest that PNNs are required for brain aromatase regulation of sIPSCs frequency in female CA1 PYR neurons.

**Aromatase regulation of synaptic inhibition and PV-IN PNNs is female-specific and independent of sex chromosomes.** Aromatase expression is differentially regulated in male and female neurons[14,40]. For this reason, we investigated aromatase regulation of male CA1 PV-INs and synaptic inhibition. We first determined if aromatase is expressed in PV-INs of male mice.

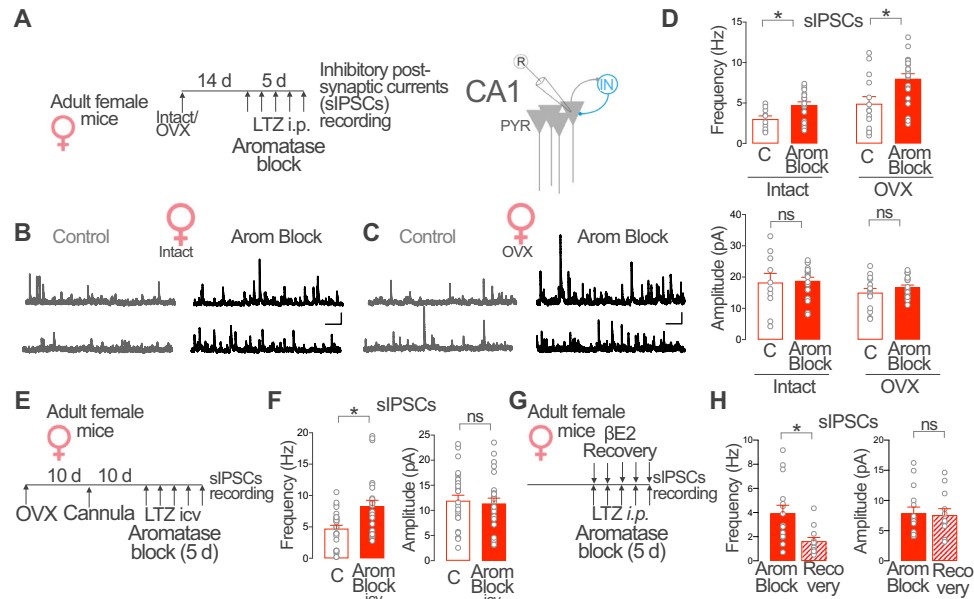

**Fig. 3 Brain aromatase regulates CA1 synaptic inhibition in female mice. A** Intact or ovariectomized (OVX) female mice received daily intraperitoneal (i.p.) injections of the aromatase blocker letrozole (LTZ) or vehicle (C) for 5 days. Spontaneous Inhibitory Post-Synaptic Currents (sIPSCs) were recorded from CA1 pyramidal (PYR) neurons. **B** Representative recordings of sIPSCs in vehicle (C) and letrozole (Arom Block) treated intact female mice. **C** Same as **B**, but in OVX female mice. Scale bar in **B** and **C**: 50 pA, 1 s. **D** Group data for experiment described in **A**. Frequency, two-way ANOVA, C/Arom Block $F_{(1, 61)} = 14.10$, $p = 0.0004$, Intact/OVX $F_{(1, 61)} = 18.17$, $p < 0.0001$. Bonferroni comparison tests, Intact C vs. Arom Block $p = 0.002$; OVX C vs. Arom Block $p = 0.002$. Amplitude, two-way ANOVA C/Arom Block $F_{(1, 61)} = 0.70$, $p = 0.41$, Intact/OVX $F_{(1, 61)} = 3.18$, $p = 0.08$. Bonferroni comparison tests, Intact C vs. Arom Block $p > 0.99$; OVX C vs. Arom Block $p > 0.99$. Intact, $n = 10$, 19 neurons from 3 mice per group; OVX, $n = 14$, 21 neurons from 3 mice. **E** Adult OVX female mice received daily intracerebroventricular (icv) injections of vehicle (C) or the aromatase blocker LTZ (Arom Block icv) for 5 days. **F** Group data for experiment described in **E**. Frequency, Two-tailed Mann–Whitney test, $U = 197$, $p = 0.006$; amplitude, unpaired two tailed $t$-test, $t_{(51)} = 0.34$, $p = 0.73$; $n = 27$, 26 cells from 5 mice per group. **G** Female mice were treated with the aromatase blocker LTZ (Arom Block) or LTZ and 17β-estradiol (βE2, Recovery) for 5 days. **H** Group data for experiment described in **G**. Frequency, Two-tailed Mann–Whitney test, $U = 33$, $p = 0.004$. Amplitude, Unpaired two tailed $t$-test, $t_{(25)} = 0.22$, $p = 0.8$; $n = 12$, 15 neurons from 3 mice per group. Graphs represent mean ± SEM (columns and bars) and individual values (gray circles). *$p < 0.05$; ns $p > 0.05$. Source data are provided as a Source Data file.

Similar to female, immunohistochemical analysis showed expression of aromatase protein in male mice CA1 PV-INs (Fig. 5A). Aromatase protein expression in PV-INs was not different in male and female mice (Fig. S4A). In order to test whether aromatase regulates male CA1 synaptic inhibition, we exploited pharmacological blockade and determined the effect on sIPSCs in CA1 PYR neurons of male mice (Fig. 5B). Surprisingly, in contrast to female mice, aromatase blockade in male mice failed to increase sIPSCs frequency or amplitude in males (Fig. 5C). We also tested the regulation of PNNs by aromatase in male mice by measuring WFA staining surrounding CA1 PV-INs. Aromatase blockade did not alter PNNs in CA1 PV-INs of male mice (Fig. 5D). These results suggest a sex difference in aromatase regulation of CA1 synaptic inhibition.

Sex differences in the brain (and in the rest of the organism) stem from gonadal (hormonal) or direct sex chromosome effects[55]. To investigate the origin of female-specific regulation of CA1 synaptic inhibition and PNNs by aromatase, we used the Four Core Genotype (FCG) mice. FCG mice have a deletion of the *Sry* gene in Y chromosome (Y$^{Sry-}$) and an autosomal *Sry* transgene, making gonadal sex determination independent of sex chromosomes[56]. Two female genotypes (XX and XY$^{Sry-}$) can be found in FCG mice, allowing the determination of the chromosomic and gonadal contributions to sex differences. We pharmacologically blocked aromatase in FCG female mice bearing XX and XY$^{Sry-}$ sex chromosomes (Fig. 5E). We observed that aromatase blockade increased CA1 PYR neuron sIPSCs frequency, but not amplitude in XX females (Fig. 5F). Aromatase blockade of female FCG mice bearing XY$^{Sry-}$ sex chromosomes

produced a similar increase in sIPSCs frequency, leaving sIPSCs amplitude unaffected (Fig. 5F). Moreover, blockade of aromatase in male FCG mice bearing female sex chromosomes (XX) failed to increase sIPSCs frequency or amplitude (Fig. S4B, C). Frequency distribution analysis of WFA staining intensities showed that aromatase blockade increases the intensity of WFA staining surrounding CA1 PV-INs in both XX and XY$^{Sry-}$ female mice (Fig. 5G). Thus, female specific aromatase effects on sIPSCs and PV-INs PNNs depend on gonadal sex but are independent of the genetic sex of the brain.

**Aromatase regulates CA1 PV-IN activity in vivo.** Aromatase regulation of CA1 synaptic inhibition in intact and OVX female mice suggests a direct impact of estradiol synthesis on the activity of PV-INs in vivo. We first tested this idea in intact female mice using fiber photometry to monitor PV-IN activity in freely moving mice exploring a familiar enclosure. Using viral vectors, we expressed the calcium indicator GCaMP6m in CA1 PV-INs and implanted an optical fiber above CA1 *stratum pyramidale* (Fig. 6A, B and Fig. S5A). We simultaneously tracked mouse position and speed changes in the enclosure and recorded calcium-dependent GCaMP6m fluorescence (Fig. 6C). We used the latter as a surrogate of PV-INs activity. Recordings were performed longitudinally, before (Control, vehicle treatment), after pharmacological blockade of aromatase (Arom Block) and after a recovery period in which animals received exogenous βE2 to compensate the lack of endogenous synthesis (Recovery, Fig. 6A).

Previous studies have shown that CA1 PV-INs activity is regulated by locomotion[31–33,57,58]. While the PV-IN population

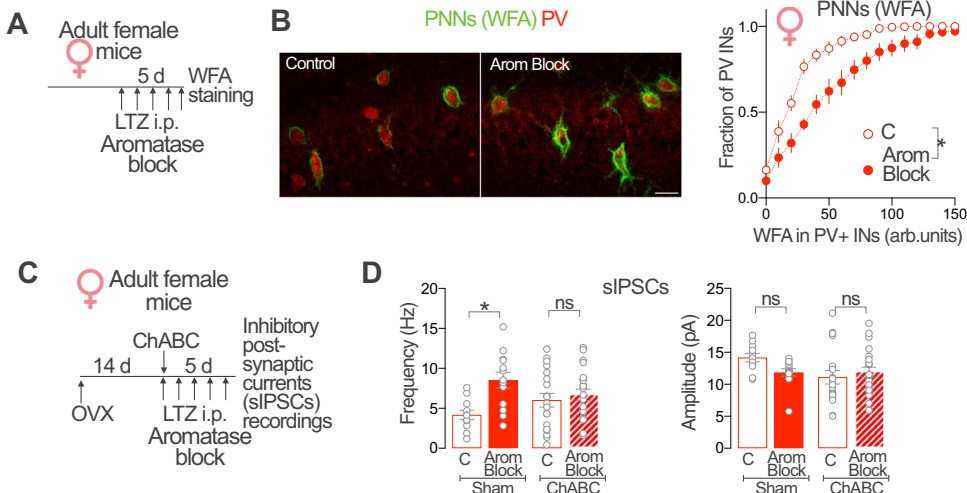

**Fig. 4 PNNs are required for extragonadal aromatase regulation of CA1 synaptic inhibition. A** Adult intact female mice received daily intraperitoneal (i.p.) injections of aromatase blocker letrozole (LTZ, Arom Block). After 5 days of treatment, perineuronal nets (PNNs) were evaluated quantifying WFA staining intensity. **B** Representative images showing WFA staining of PNNs (green) surrounding CA1 parvalbumin PV-INs (red) of vehicle (Control) and aromatase blocker letrozole (Arom Block) treated female mice. Scale bar: 20 μm. Graph shows frequency distribution analysis of WFA staining intensities around PV-INs, arbitrary units (arb. units). Aromatase blockade increases WFA staining intensity around female CA1 PV-IN. Two-way ANOVA, Treatment $F(1, 12) = 14.88$, $p = 0.0023$; $n = 6$, 8 mice. **C** Adult female mice received a single intrahippocampal injection of Chondroitinase ABC (ChABC) or vehicle (Sham) 14 days after ovariectomy (OVX). On the same day of the ChABC or vehicle intracerebral injection, mice received the first intraperitoneal (i.p.) daily injection of aromatase blocker letrozole (Arom Block) or vehicle (C). After 5 days of treatment, CA1 PYR neuron spontaneous Inhibitory Post-Synaptic Currents (sIPSCs) frequency and amplitude were determined. **D** ChABC treatment prevented the increase of sIPSCs frequency caused by Aromatase blockade (Left graph). Two-way ANOVA, Treatment Arom block $F(1, 60) = 8.432$, $p = 0.0052$; treatment ChABC, $F(1, 60) = 3.715 \times 10^{-7}$, $p = 0.9995$. Bonferroni's comparison tests, C vs. Arom Block (Sham) $p = 0.0045$, C vs. Arom Block (ChABC) $p > 0,99$. No significant changes were observed in sIPSCs amplitude (right graph). Two-way ANOVA, Treatment Arom block $F(1, 60) = 0.72$, $p = 0.4$; treatment ChABC, $F(1, 60) = 2.78$, $p = 0.1$. Bonferroni's comparison tests, C-Arom Block (Sham) $p = 0.22$, C-Arom Block (ChABC) $p = 0.97$; $n = 11$, 13, 19, 21 cells from 3 mice per group. Graphs represent mean ± SEM (symbols and bars) and individual values (gray circles). *$p < 0.05$; ns, $p > 0.05$. Source data are provided as a Source Data file.

is heterogenous, the major contribution of PV basket cells should provide an appropriate signal-to-noise ratio to detect the associated effects. Thus, we analyzed the activity of PV-INs during immobility/locomotion transitions. In control conditions, we observed an increase in the activity of PV-INs when mice started running and a decrease when mice stopped (Fig. 6D), consistent with the previously reported positive modulation of CA1 PV-INs by locomotion. To compare locomotion-related regulation of PV-IN activity, we plotted the relationship between GCaMP6m fluorescence and mice acceleration/deceleration in control, aromatase blockade sessions (Fig. 6E) and calculated an acceleration modulation index (Fig. 6F, see the "Methods" section). Despite locomotion regulated activity of PV-INs in all treatment conditions, the modulation of PV-INs activity by acceleration/deceleration was stronger in animals under aromatase blockade treatment (Arom Block) as compared with control sessions (Control, Fig. 6E, F). The relationship between PV-IN activity and acceleration reverted to control levels when animals received βE2 to compensate for lack of endogenous estradiol synthesis (Recovery, Fig. 6E, F). No change in running speed and immobility were detected in the Arom Block sessions with respect to control conditions excluding indirect effects (Fig. S5B).

These results show that aromatase blockade increases coupling of the activity of CA1 PV-INs to locomotion in female mice and suggest that aromatase-derived estradiol regulates the activity of PV-INs in vivo.

**Brain aromatase regulates hippocampal network activity**. PV-INs organize different forms of behaviorally relevant hippocampal network activity, including gamma and theta oscillations and sharp-wave ripples (SWRs)[34,35]. Aromatase regulation of PV-IN activity suggests that local estradiol synthesis may affect these

forms of hippocampal network activity in vivo. We choose to evaluate the impact of aromatase blockade on SWR, gamma and theta oscillations because of the critical dependency on local CA1 PV-INs[12,59]. To this purpose, we performed longitudinal local field potential (LFP) recordings with linear silicon probes in the dorsal CA1 region of head-fixed OVX female mice trained to rest in a wheel. Recordings were performed on the fifth day of treatment with vehicle (Control), the aromatase blocker letrozole (Arom Block) or after simultaneous treatment with aromatase blocker and βE2 (Recovery, Fig. 7A) 90 min after the last injection. The use of OVX female mice allowed us to test the in vivo effect of aromatase blockade on CA1 activity and determine the impact of βE2 recovery treatment in the absence of gonadal synthesis.

During periods of relaxed immobility, SWRs were recorded from the *strata pyramidale* (SP) and *radiatum* (SR) of OVX female mice in all treatment conditions (Fig. 7B). Aromatase blockade significantly increased the occurrence of SWRs, which exhibited lower ripple power (both at the mean and maximum values, Fig. 7C, D). To exclude potential effect of detection thresholds, we concatenated recordings obtained before and after aromatase blockade to use a common threshold (3 × standard deviation, SD), and found consistent increase of SWR rate in the Arom Block group ($Z = -1.888$, $p = 0.0295$. Wilcoxon signed rank test, $n = 5$ recordings from 3 animals). We neither found effect of vehicle injection (before vs. after) in either SWR rate, mean and maximum power (Fig. S6A). Instead, the mean and maximum ripple power reverted to control values after the recovery treatment (Fig. 7D), supporting effect specificity.

To gain further insights, gamma (30–90 Hz) and theta oscillations (4–10 Hz) were examined during periods of active movement in the wheel. Aromatase blockade produced a decrease

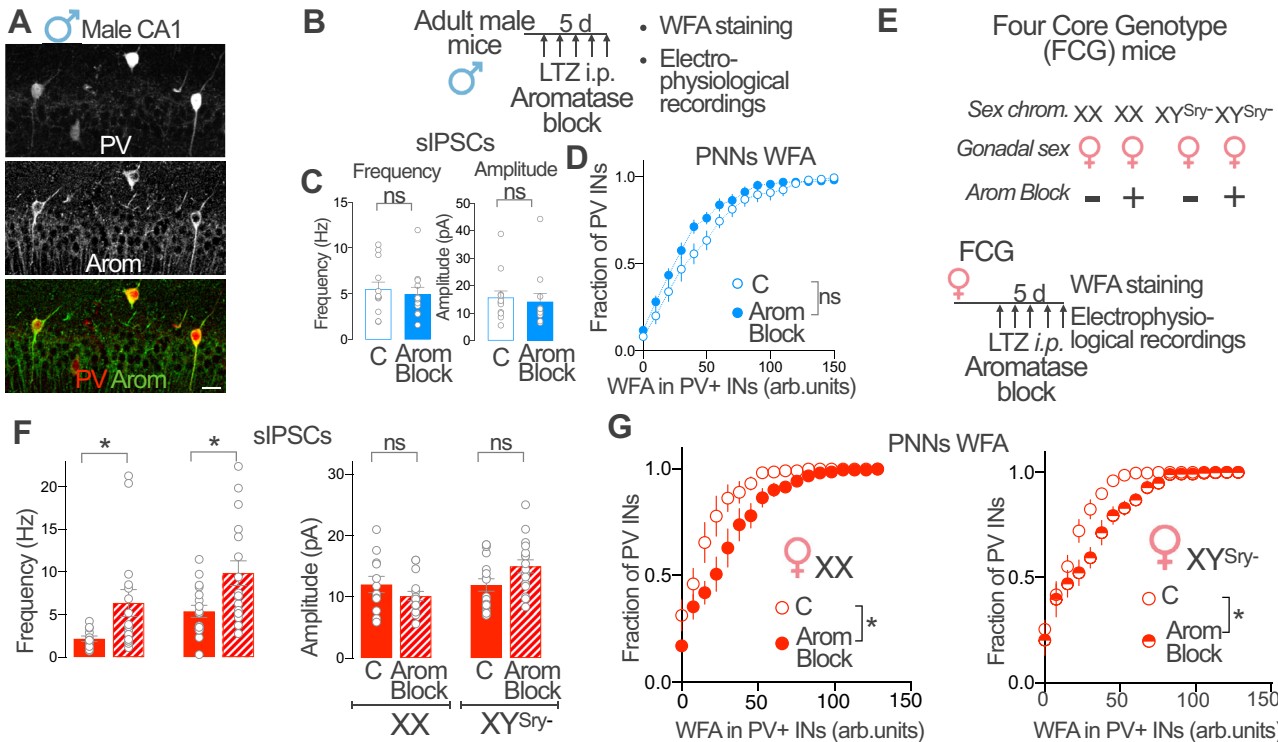

**Fig. 5 Aromatase regulation of synaptic inhibition and PV-IN PNNs is female-specific and independent of sex chromosomes. A** Aromatase protein (Arom, red) and parvalbumin (PV, green) expression in male mice dorsal CA1. Single-channel images are represented in gray scale. Scale bar: 20 μm. **B** Male mice received intraperitoneal (i.p.) injections of the aromatase blocker letrozole (LTZ) or vehicle (C) and processed for perineuronal nets (PNNs) analysis or electrophysiological recordings. **C** Group data for experiment in **B** (sIPSCs). Two-tailed Mann–Whitney tests, frequency $U = 62$, $p = 0.41$; amplitude, $U = 58$, $p = 0.29$; $n = 13$, 12 cells, 3 mice per group. **D** Group data for experiment in B (WFA staining). Two-way ANOVA, Treatment $F_{(1, 13)} = 1.897$, $p = 0.1916$; $n = 7$, 8 mice. **E** XX or XY$^{Sry}$ female mice were treated with LTZ and processed for PNNs analysis or electrophysiological recordings. **F** Group data for experiment in **E** (sIPSCs). Two-way ANOVA, Frequency Arom Block $F_{(1, 58)} = 12.68$, $p = 0.0007$, Chromosomes $F_{(1, 58)} = 7.592$, $p = 0.0078$, interaction $F_{(1, 58)} = 0.01428$, $p = 0.9053$. Bonferroni's comparisons, C vs Arom Block (XX), $p = 0.0486$, C vs Arom Block (XY$^{Sry−}$) $p = 0.0157$. Amplitude, Treatment $F_{(1, 58)} = 0.3$, $p = 0.58$, Chromosomes $F_{(1, 58)} = 5.17$, $p = 0.03$, interaction $F_{(1, 58)} = 5.17$, $p = 0.02$. Bonferroni's comparisons, C-Arom Block (XX), $p = 0.4$, C vs. Arom Block (XY$^{Sry−}$) $p = 0.07$; $n = 12$, 16, 17, 17 cells, 3 mice per group. **G** Group data for experiment in **E** (WFA staining). Two-way ANOVA, Female XX Arom Block $F_{(1, 8)} = 5.72$, $p = 0.0438$, interaction $F_{(17, 136)} = 3.15$, $p = 0.0001$; $n = 5$ mice. Female XY$^{Sry−}$, Arom Block $F_{(1, 8)} = 13.6$, $p = 0.006$, interaction $F_{(17, 136)} = 3.29$, $p < 0.0001$; $n = 6$, 4 mice. Graphs represent mean ± SEM. In **C** and **F**, circles represent individual values. $*p < 0.05$; ns, $p > 0.05$. Source data are provided as a Source Data file.

in the power of both theta and gamma oscillations that reverted to control levels after the recovery treatment (Fig. 7E). Similar changes were identified for the slow (30–60 Hz) and fast (60–90 Hz) bands (Slow gamma: Chi-square = 18.028, $p = 0.00012$; Fast gamma: chi-square = 23.752, $p < 0.0001$. Kruskal–Wallis test, $n = 11$, 16, 13 for Control, Arom Block and Recovery groups, respectively). The ratio of 8 and 4 Hz theta oscillation power remained constant in all experimental conditions tested (Fig. 7E; see also Fig. S6B for vehicle experiments).

Therefore, aromatase blockade has an impact on OVX female CA1 hippocampal network activity dependent on CA1 PV-INs that is reversed by exogenous βE2 administration. These data suggest that brain aromatase regulates behavioral relevant hippocampal activity in female mice.

**Brain aromatase regulates hippocampal memory**. The impact of aromatase blockade in hippocampal oscillations recorded in head-fixed female mice prompted us to investigate the consequences of suppressing brain estradiol synthesis on a hippocampal dependent memory task in freely behaving mice. With this purpose, we compared the performance of OVX female mice that carried on the novel object location task (NOL, Fig. 8A) on the fifth day of systemic treatment with vehicle (Control), the aromatase blocker letrozole (Arom Block) or the aromatase

blocker and βE2 (Recovery, Fig. 8B). OVX female mice treated with vehicle showed increased preference for the displaced object during the test session (Fig. 8B, Control). This preference was not observed in OVX female mice treated with pharmacological aromatase blocker (Fig. 8B, Arom Block). The preference for the displaced object was also observed in the recovery group (Recovery, Fig. 8B). Finally, we evaluated the effect of aromatase blockade in male mice (Fig. 8C). Male mice treated with aromatase blocker showed preference for the displaced object that was similar to male mice treated with vehicle (Fig. 8C). We did not observed effects of the different treatments during the familiarization session in the total distance traveled, object exploration or object preference (Fig. S7), excluding nonspecific effects.

Altogether, these results suggest that brain aromatase is required for object location memory in female mice. On the other hand, male performance was unaffected, suggesting different dependency on estrogen synthesis of male and female object location memory.

## Discussion

Our results show that female CA1 PV basket cells, together with CA1 PYR neurons, are a source of estradiol in the hippocampus. Brain-derived estradiol negatively regulates synaptic inhibition in

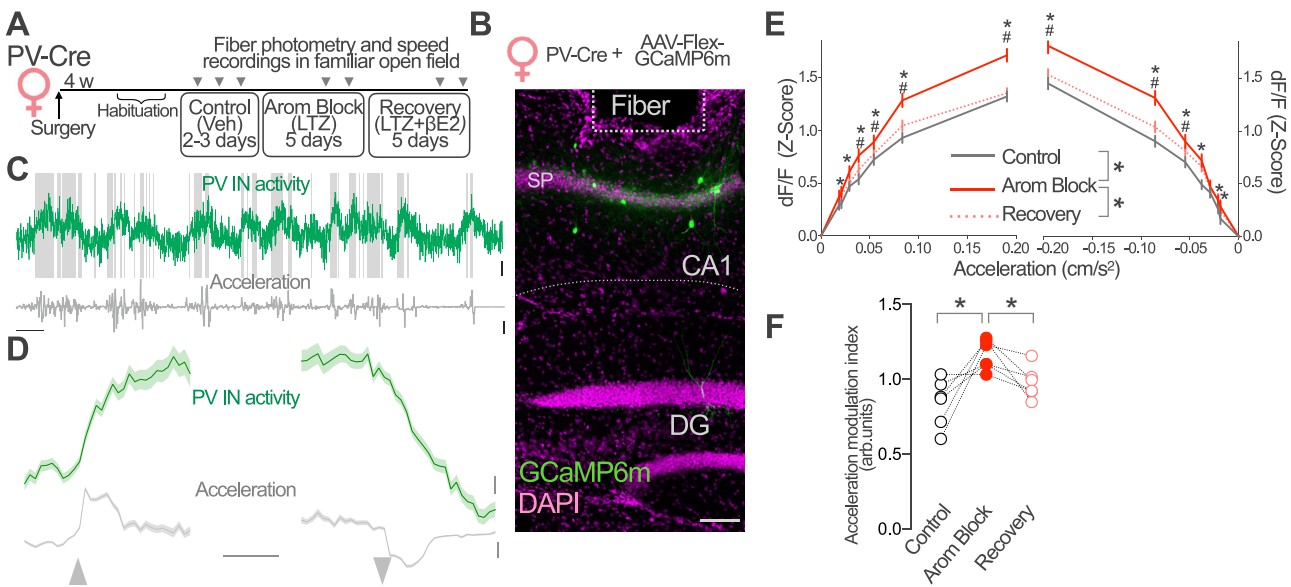

**Fig. 6 Aromatase regulates PV-IN activity in vivo. A** Parvalbumin-Cre (PV-Cre) female mice were infected with AAV-Flex-GCaMP6m and an optic fiber was implanted above CA1 area. After habituation to an open field arena (5 days), mice received daily intraperitoneal injections of vehicle (Control, 2–3 days), the aromatase blocker letrozole (LTZ, Arom Block, 5 days) and LTZ+ 17β-estradiol (βE2, Recovery). Fiber photometry signal and speed were registered 90 min after the last injection during each treatment condition as indicated, while animals were freely exploring the open field (10 min). **B** Representative image of GCaMP6m expression (green) and optic fiber placement above CA1 *stratum pyramidale* (SP). Scale bar 0.1 mm. **C** Representative GCaMP6m fluorescence (green) during exploration in control conditions. Shaded areas mark mobility periods. Instantaneous acceleration (gray) was used to analyze locomotory behavior. Scale bars: 10 s, 1% dF/F and 0.1 m/s². **D** Event-triggered average traces for immobility to locomotion (left) and locomotion to immobility (right) transitions. Traces show mean ± SEM for all recorded mice (n = 6) during control sessions. Scale bars: PV IN activity, 0.1 Z-score; acceleration 0.02 m/s;² 1 s. **E** Plots show PV-INs response to acceleration (right) and deceleration (left) in each treatment condition. PV-INs responses increased during Arom Block treatment (red) with respect to control (gray) and recovery (dotted red) conditions. Two-way, repeated measures ANOVA, treatment $F(2, 238820) = 246.8$, $p < 0.0001$. Bonferroni's comparison tests, *$p < 0.05$ Control vs. Arom Block, #$p < 0.05$ Recovery vs. Arom Block. n corresponds to instant measurements, data obtained from 6 female mice. Data represent mean ±95% confidence interval. **F** The acceleration modulation of PV-INs increased after aromatase pharmacological blockade (Arom Block) and returned to control levels when animals receive βE2 to compensate for lack of endogenous estradiol synthesis (Recovery). One-way ANOVA, $F(2, 15) = 9.94$, $p = 0.002$; Bonferroni's comparison tests Control vs. Arom Block $p = 0.001$, Arom Block vs. Recovery $p = 0.038$; $n = 6$ mice. Data represents values for individual mice. *$p < 0.05$. Source data are provided as a Source Data file.

CA1 PYR neurons. Although the precise cellular origin of estradiol remains unknown, this process involves suppression of PNN function specifically in PV-INs. Consistent with this, reducing estradiol synthesis has a direct impact on PV-IN activity in vivo: aromatase blockade influences the coupling of PV-IN activity to locomotion, the dynamics of hippocampal oscillations and object location memory in female mice. Aromatase regulation of CA1 inhibition and PV-INs is only observed in female mice, depends on gonadal sex and is independent of sex chromosomes. These results suggest that PV-INs mediate a mechanism by which brain-derived estradiol could regulate brain activity involved in information processing and storage in a sex-specific manner.

Previous reports show that aromatase mRNA and protein are expressed in CA1 neurons[13,39]. Our results confirm and extend these observations by showing the presence of aromatase mRNA and immunoreactivity in PV-INs of this hippocampal region. The expression of aromatase and the accumulation of βE2 suggest that PV-INs are a source of estradiol. In addition to modulate excitatory drive in the hippocampus by direct action on excitatory neurons and synapses[60–62], estradiol regulates the function of PV-INs, which are critical in organizing hippocampal network activity[12]. In this way, PV-INs may act as sources and targets of estradiol in the female hippocampus. Estradiol may simultaneously increase hippocampal excitatory synaptic function and decrease synaptic inhibition, through direct action on PV-INs, promoting in this way plasticity of excitatory neuronal network in females. Akin to the results presented here, aromatase blockade

regulates CA3-CA1 synaptic plasticity exclusively in female mice[49].

What sources of estradiol may be responsible for regulating functional CA1 synaptic inhibition and PV-INs in female mice? INs and inhibitory synapse activity are regulated by estrous cycle and pubertal-dependent hormonal levels fluctuations in various brain regions, suggesting a role for plasma derived estradiol in the regulation of inhibitory neurotransmission[18–20]. However, systemic and intracerebroventricular administration of an aromatase blocker increased synaptic inhibition in mice lacking the ovaries, the main source of plasma estradiol. This suggests a role for extragonadal, and in particular, brain derived estradiol, in regulating hippocampal inhibition. Experiments in OVX mice demonstrate that brain aromatase regulates CA1 synaptic inhibition and PV-IN function independently of estrous cycle regulation of brain estradiol synthesis[63]. Alternative approaches to the systemic aromatase blockade in OVX mice used in our experiments would be necessary to unveil the precise identity of aromatase expressing cells that regulate PV-INs and CA1 synaptic inhibition. Nonetheless, our experiments suggest that aromatase expressed in hippocampal PV-INs or PYR neurons could act as a local source of estradiol to regulate CA1 inhibition through paracrine or autocrine actions onto PV-INs. Our results do not exclude a role for circulating estradiol in regulating CA1 inhibition. Indeed, we observed that exogenous applied βE2 regulates CA1 synaptic inhibition in the absence of endogenous estradiol synthesis, suggesting a cooperation between

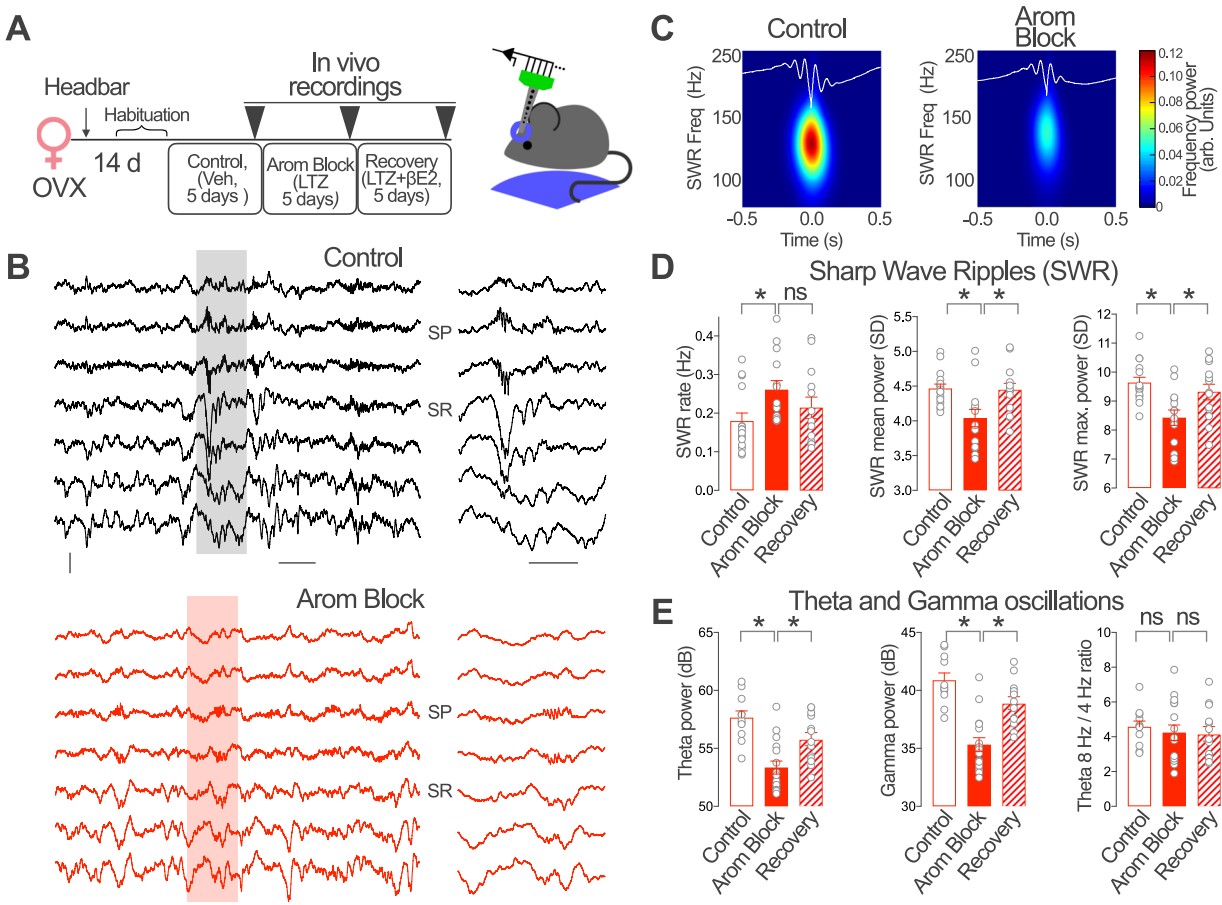

**Fig. 7 Brain aromatase regulates the dynamics of SWRs and hippocampal oscillations in awake female mice. A** Ovariectomized (OVX) female mice received daily intraperitoneal injections of vehicle (Control), the aromatase blocker letrozole (LTZ, Arom Block) and LTZ + βE2 (Recovery). Recordings were performed on the 5th day of consecutive treatments. **B** Representative recordings from an OVX female in Control and Arom Block conditions. One Sharp Wave Ripple (SWR) event is shown at right at enlarged time scale. Scale bars: 0.1 mV, 0.2 s (left), 0.1 s (right). SP *stratum pyramidale*, SR *stratum radiatum*. **C** Mean SWR events from recordings after vehicle (Control) or LTZ (Arom Block) treatment. **D** Group statistic effects for SWRs. SWR Rate, Kruskall–Wallis test, $H = 7.99$, $p = 0.018$, Dunn´s multiple comparisons, C vs. Arom Block $p = 0.01$, Arom Block vs. Recovery $p = 0.18$. One-way ANOVA, SWR mean power, $F_{(2, 39)} = 5.886$, $p = 0.006$; Bonferroni's comparisons, C vs. Arom Block $p = 0.007$, Arom Block vs. Recovery $p = 0.015$. SWR max power, $F_{(2, 39)} = 7.028$, $p = 0.003$; Bonferroni's comparisons, C vs. Arom Block $p = 0.001$, Arom Block vs Recovery $p = 0.03$; $n = 15$, 14, 13 recordings from 4 Control, 4 Arom Block and 4 Recovery treated animals, respectively. **E** Group statistic effects for theta and gamma oscillations. One-way ANOVA, Theta power, $F_{(2, 37)} = 13.99$, $p < 0.0001$, Bonferroni's comparisons, C vs. Arom Block $p < 0.0001$, Arom Block vs. Recovery $p = 0.007$. Gamma power, $F_{(2, 37)} = 22.54$, $p < 0.0001$, Bonferroni's comparisons, C vs. Arom Block $p < 0.0001$, Arom Block vs. Recovery $p = 0.0002$. Theta 8 Hz/4 Hz ratio power, $F_{(2, 37)} = 0.2265$, Bonferroni's comparisons, C vs. Arom Block $p > 0.99$, Arom Block vs. Recovery $p > 0.99$; $n = 11$, 16, 13 recordings from 5 Control, 5 Arom Block and 4 recovery treated animals, respectively. Graphs represent mean ± SEM (columns and bars) and individual values (gray circles). *$p < 0.05$; ns $p > 0.05$. Source data are provided as a Source Data file.

brain-derived and peripheral estrogens[64]. Our experiments indicate that brain aromatase activity is not required for exogenous βE2 to increase CA1 synaptic inhibition, in accordance with the independent regulation of brain and ovarian βE2 synthesis.

PV-INs show a remarkable level of functional plasticity[53,65]. PNNs have emerged as critical regulators of the excitability and network activity of neocortical[66] and hippocampal PV-INs[44,54]. In addition, in different brain regions, including the hippocampus, PNNs show sex differences and are regulated by sex hormones[67,68]. Our results suggest that aromatase negatively regulates PNNs in female CA1 PV-INs through estradiol production. Estrogen receptors (ER) are abundantly expressed in hippocampal excitatory and inhibitory neurons, including PV-INs[19,69]. ER activation promotes synaptic plasticity and improves hippocampal-dependent memory[11,70,71], a function that is also

associated to PNNs[53,72,73]. While the link between ER activation and PNNs remains to be firmly stablished, it may involve ER regulation of gene expression in hippocampal neurons[74]. Interestingly, ChABC experiment links PNNs to aromatase regulation of synaptic inhibition in CA1 PYR neurons, suggesting that PNNs are required for estradiol regulation of synaptic inhibition. Reducing estradiol production by aromatase blockade increases the intensity of PNNs and may promote in this way PV-IN activity. This result is consistent with the role of PNNs in increasing excitability of PV-INs in the hippocampus and with a presynaptic origin of the observed increased on sIPSCs frequency after aromatase blockade. PNNs promote excitability of PV-INs by increasing glutamatergic input and by regulating membrane proteins, such as voltage-gated potassium channels, that influence intrinsic excitability in this IN subtype[54,75]. However, the lack of

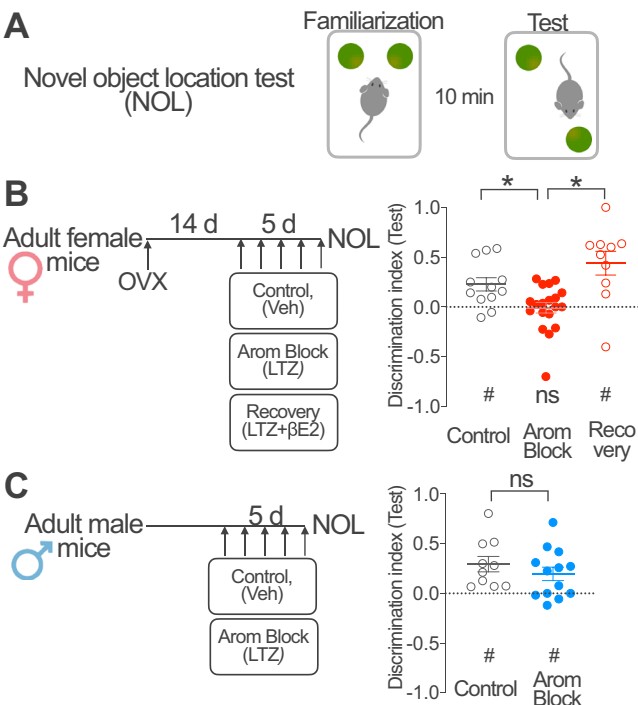

**Fig. 8 Brain aromatase regulates hippocampal memory. A** The cognitive effect of aromatase blockade and βE2 recovery was studied using the novel object location test (NOL) with 10 min inter-trial interval between the familiarization and test sessions. ∫ **B** Fourteen days after ovariectomy (OVX), adult female mice received daily intraperitoneal (i.p.) injections of vehicle (Control), the aromatase blocker letrozole (LTZ, Arom Block) or the aromatase blocker LTZ+ βE2 (Recovery) for 5 days. Mice performed the NOL test 90 min after the last injection. Graph represents population data. Discrimination index > 0 indicates preferential exploration of the displaced object during the test session. Dotted line represents chance levels. One sample t test (indicated below the graph): Control, $t(11) = 3.41$, #$p = 0.006$, Arom Block, $t(18) = 0.14$, n.s. $p = 0.89$; Recovery, $t(9) = 3.67$, #$p = 0.005$. One-way ANOVA (indicated above the graph): $F(2, 38) = 9.12$, $p = 0.0006$; Bonferroni's comparison tests, C vs. Arom Block $p = 0.03$, Arom Block vs. Recovery $p = 0.0003$; $n = 12, 19, 10$ mice. **C** Adult male mice received daily intraperitoneal injections of vehicle (Control) or the aromatase blocker letrozole (Arom Block) for 5 days. Mice performed the NOL test 90 min after the last injection. Graph represents population data. Discrimination index > 0 indicates preferential exploration of the displaced object during the test session. Dotted line represents chance levels. One sample $t$ test (indicated below the graph): Control, $t(9) = 3.82$, #$p = 0.004$; Arom Block, $t(12) = 2.89$, #$p = 0.013$. Unpaired two-tailed $t$ test (indicated above the graph), $t(21) = 0.96$, n.s. $p = 0.35$; $n = 10, 13$ mice per group. Graphs represent mean ± SEM (line and bars) and individual values (circles) for each experimental condition. * and #$p < 0.05$; n.s. $p > 0.05$. Source data are provided as a Source Data file.

effect of ChABC treatment on sIPSCs frequency in untreated OVX mice suggests that additional mechanisms may be involved in PNNs-mediated control of PV-INs activity. The unaltered sIPSCs amplitude observed after aromatase blockade is suggestive of no major alteration of mechanisms regulating post-synaptic GABA A receptor function[76].

Our results point to a differential contribution of specific subtypes of PV-INs to hippocampal estradiol synthesis. Aromatase expression is higher in PNN+ and SATB1+ PV-INs as compared with PNN− and SATB1− PV-INs. Since PNNs surround mostly PV basket cells[45], that also express SATB1[45,46], and to a much lesser extent bistratified cells, axoaxonic cells and

O-LM neurons, our data identify PV basket cells as a major source of estradiol among PV-INs. Moreover, the involvement of PNNs in aromatase blockade regulation of sIPSCs frequency in CA1 PYR neurons suggest that a neuronal population enwrapped with PNNs, namely PV basket cells, mediates aromatase effects on CA1 synaptic inhibition. Recent evidence suggests that peripheral estradiol modulates somatostatin-expressing O-LM interneurons[77]. Thus, different mechanism triggered by estradiol may alter sources of inhibition targeting somatic and dendritic neuronal compartments.

We observed regulation of CA1 synaptic inhibition, PNNs and novel object location memory by aromatase blockade in female but not in intact male mice. The mechanisms underlying this sex effect may include different post-translational control of aromatase activity through phosphorylation in male and female PV-INs[78] and different estrogenic effects on male and female neurons[79,80]. The brain regulates the local concentration of estradiol independently from the periphery[81–83]. Estradiol production in the hippocampus has been shown to increase upon kainic acid treatment in male and female rats[84], suggesting activity-dependent regulation of estradiol synthesis. Indeed, aromatase inhibition in male and female rats reduced kainic acid-induced seizures[84], as would be expected from an enhanced synaptic inhibition as a consequence of reduced estradiol synthesis. It remains to be tested if estradiol regulates male PV-INs under specific activity conditions, in physiologic or pathologic situations. Brain aromatase has been shown to regulate hippocampal memory in male and female mice[4,7,14]. Memory impairment caused by genetic or pharmacological aromatase blockade may be dependent on the relative impact of these strategies on aromatase activity in the circuits and cell types involved on specific memory processes. Cell type-specific strategies focused on circuits and cell types involved in the acquisition, consolidation or retrieval of specific forms of memory will accurately describe sex effects of estrogen synthesis in memory.

Although sex differences in hippocampal inhibitory activity have been previously described[85], the impact of sex chromosomes (XX, XY) and gonadal secretions on sex effects in the hippocampal inhibitory system is poorly characterized. The analysis in FCG mice indicates that female-specific aromatase regulation of CA1 synaptic inhibition and PV-IN PNNs is determined by gonadal sex and is independent of the sex chromosomes. Gonadal secretions acting during the postnatal or prepubertal periods[20,86,87] may be responsible for setting the sex effect in aromatase regulation of synaptic inhibition.

A principal feature of CA1 PV-INs is the positive modulation of activity by locomotion[31–33]. Our in vivo fiber photometry experiments are in line with these findings and show increased activity upon locomotion start and a positive relation with female mice acceleration. Interestingly, our data suggest that female CA1 PV-IN coupling to locomotion is not rigid and can be modulated. In line with the prominent plasticity of PV-IN population, blockade of estradiol synthesis increases the response of PV-INs to changes in animal speed. This suggests that local estradiol synthesis may limit the coupling of PV-INs to locomotion by reducing PV-IN responses, in line with the results observed in the sIPSCs recordings. Reduction in estradiol synthesis may enhance synaptic and intrinsic excitability as a consequence of PNNs modifications and promote the recruitment of PV-INs by inputs activated during locomotion[88,89]. Alternatively, aromatase blockade may suppress inhibition onto PV-INs arising from other IN subtypes active during mobility[31,32]. Previous reports suggest that concentrations compatible with local hippocampal levels but several fold higher than plasma estradiol levels, regulate synaptic release from CA1 GABAergic neurons[21]. Interestingly, the mechanism is female specific and affect a group of INs that are

sensitive to cannabinoid retrograde signaling and innervate PV-INs. The phenomenon is rapid (minutes) and postsynaptic[21], in contrast with our observations. Since cannabinoid sensitive and PV-INs differentially regulate spike timing activity of PYR neurons[90] and are differently engaged by locomotion and SWRs[32], local estradiol regulation of CA1 INs may impact an extended repertoire of neuronal computations and compartment-specific synaptic inhibition in PYR neurons during different behavioral states.

We found that the dynamics of SWRs and the power of gamma and theta oscillations were altered in OVX females treated with aromatase blockers. They exhibited lower amplitude and more frequent SWRs and reduced oscillatory power. SWRs recorded in CA1 are initiated by CA3 inputs driving complex local interplay between CA1 PYR and INs[59]. Similarly, gamma oscillations, specially the slow gamma rhythm (30–60 Hz) is associated to the activity of PV basket cells[91]. Our in vitro data point to deregulation of PV-IN-PYR microcircuit function as a major contributor. However, since treatment was systemic, we cannot exclude that both local and distal effects are influencing SWRs and oscillatory activity. While the direction of the observed changes cannot be explained by simple mechanisms, the contribution of diverse populations of PYR and INs may provide additional hints. For example, deep and superficial CA1 PYR cells likely recruit different IN subtypes during SWRs[92]. Independent on the specific mechanism, decreased inhibition caused by aromatase activity blockade and the alteration of PV-IN function both have impact in strikingly distinct forms of network activation in the hippocampus, i.e. SWRs, theta and gamma oscillations. The prominent role of oscillations in temporal coding and information binding, together with the function of SWRs in memory consolidation[36] and the reduced network plasticity associated with high PV-IN activity[53], all suggest that cognitive deficits caused by aromatase inhibitors in humans and mice may arise from the impact of local estradiol synthesis on PV-INs.

In conclusion, our results are a first description of a sex-specific control of hippocampal inhibition by brain estradiol synthesis. INs should be considered to fully understand the hormonal regulation of the brain and in particular, sex hormone-dependent control of hippocampal function and plasticity.

## Methods

**Animals.** Experiments were performed according to protocols approved by the Institutional Animal Care and Use Committee of the Cajal Institute and by local veterinary authorities (Comunidad de Madrid). Group housed adult C57BL/6J wild type, PV-Cre (*Pvalb tm1(cre)Arbr/J*, a kind gift from G. Perea, Cajal Institute, Madrid) and Four Core Genotype mice (FCG, B6.Cg-Tg(Sry)2Ei *Srydl1Rlb/*ArnoJ, a kind gift from A.P. Arnold, UCLA, USA) were maintained in a 12 h light/dark cycle, 20–22 °C, 45–65% humidity and with unlimited access to food. All animals were obtained from the animal facility of the Cajal Institute. PV-Cre and FCG mice were maintained in a C57BL/6J and CD1 genetic background, respectively. Animals were sacrificed at 10–12 weeks for histological analysis and ex-vivo electrophysiological experiments. Fiber photometry and in vivo electrophysiology were performed at 11–13 weeks of age. Behavioral experiments were performed at 12–15 weeks of age. Genotyping of FCG mice was performed by RT-PCR detection of *Sry* and *Ssty* (located in the Y chromosome) gene transcripts. The estrous cycle in intact female mice was not monitored.

**Reagents and AAVs.** Letrozole (Tocris) was dissolved in DMSO to 12.5 mg/ml, further dissolved in saline solution to 62.5 µg/ml and administered at a dose of 0.5 mg/kg in intraperitoneal (i.p.) injections of 8 ml/kg. For intracerebroventricular (icv) administration, letrozole was dissolved in saline and 1% DMSO to a concentration of 0.5 µg/µl. A total volume of 1 µl was infused unilaterally at a rate of 100 nl/min.

17β-estradiol (Sigma) was dissolved in DMSO and further dissolved in saline and injected at a dose of 2 mg/kg. Mice were sacrificed or the experiments started 90 min after the last ip or icv treatment administration. Chondroitinase ABC, from Proteus Vulgaris (Sigma) was dissolved to 100 units/ml in PBS with 0.01% BSA. The day of the surgery, this stock was diluted in PBS with 0.02% BSA to a final concentration of 40 units/ml before being injected in the hippocampus following the procedure described below.

Adeno associated viruses (AAVs) used in this study were produced by Addgene (pAAV.Syn.Flex.GCaMP6m.WPRE.SV40, serotype 9) and the University of North Carolina vector core (AAV-EF1a-DIO-EYFP-WPRE-pA, serotype 5).

**Surgery.** Analgesic treatment (paracetamol 0.2 g/kg) was administered for 4 days around surgery. Anesthesia was induced at 5% and maintained at 1.5–2.0% isoflurane (w/v). Mice were placed in a stereotaxic frame (RWD) and craniotomies were performed using stereotaxic coordinates adapted from a mouse brain atlas to target the dorsal CA1: −2.3 anterior–posterior; ±1.65 medial–lateral; −1,6 dorsal–ventral. Injections of AAV or ChABC (0.5–0.6 µl) were performed using graduated pipettes (Drummond Scientific Company), broken back to a tip diameter of 10–15 µm, at an infusion rate of ~0.05 µl min−1. Micropipettes were left in place 5 min following microinjection and slowly retracted (0.4 mm/minute) to avoid reflux of the viral solution. Experiments involving AAVs started on the 4th week after the viral injection. Letrozole treatment in ChABC/Vehicle injected mice started immediately after intracranial surgery. For intracerebroventricular administration, a guide cannula was implanted with the tip aiming the left lateral ventricle (coordinates: −0.2 anterior–posterior; 0.9 medial–lateral; −2.4 dorsal–ventral), fixed to the skull as described below for the optical implants and protected with a dummy cannula. Cannula tip position was verified histologically at the end of the experiments. When indicated, 9–12 weeks-old animals were bilaterally ovariectomized under isoflurane anesthesia. Ovaries were removed by performing incisions in the lateral part of the trunk that was sutured before letting animals to recover. The experiments started 10–14 days later, as indicated in the experimental schema and figure legends.

**Immunohistochemistry and in situ mRNA detection.** Mice were injected with a lethal dose of pentobarbital (150 mg/kg) and perfused transcardiacally with cold PBS and 3–4% paraformaldehyde solution. Brains were extracted and submerged in fixative for 4 h at 4 °C. For mRNA analysis, 30 µm thick sections containing dorsal hippocampus were cut in a vibratome and immediately processed. Slices were mounted onto SuperFrost Plus microscope slides (10149870, ThermoFisher Scientific). RNAScope Assay (Advanced Cell Diagnostics) was carried out in samples from 2 female mice according to manufacturer's protocols. Briefly, sections were dehydrated at 60 °C, pretreated with a target retrieval solution and protease III, and hybridized with a probe designed to detect mouse *cyp19A1* gene (Accession No. NM_007810.3, directed towards a target region encompassing nucleotides 719–1985). Signal was amplified before revealing probe binding with Fast Red Amplification reactive (RNAscope Fast Red Detection reagents, ACD). Positive control (*ubc*, Accession No NM_019639.4) and a negative control probes (*dapB*, from *Bacillus subtilis*, Accession No. EF191515) were used to establish unspecific labeling.

For immunohistochemistry, coronal 40 µm-thick sections containing dorsal hippocampus were blocked in PBS 0.3% BSA, 5% normal goat serum (NGS) and 0.3% Triton X-100 followed by overnight incubation in PBS, 5% NGS and 0.3% Triton X-100 with primary antibody: parvalbumin (1:2000 mouse monoclonal, code 235 and 1:2000 guinea pig polyclonal, code GP42, both from Swant), aromatase (1:1000, in-house production, described and validated in ref.[93]), SATB1 (1:1000 mouse monoclonal, C-6, Santa Cruz Biotechnologies) and SST (1:250 rat monoclonal, code MAB354, Millipore). Biotinylated Wisteria Floribunda Lectin (1:500 Vector Laboratories) was incubated in the same conditions as primary antibodies. After 3 × 15 min wash in PBST at room temperature, slices were incubated with 1:500 Alexa-conjugated secondary antibodies and Streptavidin (Alexa-Fluor 488, 555, 645, Abcam) to reveal primary antibodies and biotynilated WFA, respectively. After 3 more steps of washing in PBST, slices were mounted and covered on microscope slides using DAPI containing mounting medium.

**Image analysis.** Images were obtained with a Leica SP5 confocal microscope (LEICA LAS AF software) using ×20 or ×40 objectives and 405, 488, 561 and 633 nm laser excitation wavelengths. 1024 × 1024 images with a resolution of 1.3–2.6 pixel/µm, at 3–4 µm step size were collected. Manually depicted ROIs delimiting CA1 PV, SST or pyramidal neurons were used to determine fluorescence intensity in other channels (aromatase, SATB1). To quantify aromatase mRNA, Fast Red signal dots surface density in PV-INs, pyramidal cell layer and Medial Amygdala neurons was determined in thresholded images using Fiji. Chance colocalization was determined using the same method after relative flipping of the image corresponding to aromatase mRNA. For quantification of WFA staining, a lineal ROI surrounding PV-INs (3.8 µm width) was used. Mean pixel intensity in closed and lineal ROIs was determined in equally thresholded images. Cumulative distribution was obtained for aromatase protein and WFA staining intensities from all PV, SST or pyramidal neurons analyzed in at least 3 slices from individual mice. The cumulative distributions for each individual mouse were then averaged to obtain values used for plots and to perform statistical analysis.

**Slice electrophysiology.** Acute slices for electrophysiological recordings were prepared from 10 to 13 weeks old mice 90 min after the last treatment administration. Brains were quickly removed and coronal slices (300 µm) containing the dorsal hippocampus were cut with a vibratome (4 °C) in a solution containing: 234 mM sucrose, 11 mM glucose, 26 mM NaHCO$_3$, 2.5 mM KCl, 1.25 mM

$NaH_2PO_4$, 10 mM $MgSO_4$, and 0.5 mM $CaCl_2$ (equilibrated with 95% $O_2$–5% $CO_2$). Recordings were obtained at 30–32 °C from CA1 *stratum pyramidale* neurons (pyramidal neurons, PYR) visually identified using infrared video microscopy in oxygenated artificial cerebrospinal fluid containing 126 mM NaCl, 26 mM $NaHCO_3$, 2.5 mM KCl, 1.25 mM $NaH_2PO_4$, 2 mM $MgSO_4$, 2 mM $CaCl_2$, and 10 mM glucose (pH 7.4). Patch-clamp electrodes contained intracellular solution composed of: 127 mM Cesium methanesulfonate, 2 mM CsCl, 10 mM HEPES, 5 mM EGTA, 4 mM MgATP, and 4 mM QX-314 bromide, pH 7.3 adjusted with CsOH (290 mOsm). GABA receptor-mediated spontaneous Inhibitory Post-Synaptic Currents (sIPSCs) and TTX (Latoxan, France) insensitive currents (mIPSCs) were registered by clamping neurons at 0 mV. Signals were amplified using a Multiclamp 700B patch-clamp amplifier and digitized using a Digidata 1550B (Axon Instruments, USA), sampled at 20 kHz, filtered at 10 kHz, and stored on a PC using Clampex 10.7 (Axon Instruments). Series resistance was monitored by a voltage pulse in every recorded cell and compared between experimental groups to discard effects due to recording conditions. IPSC were analyzed using pClamp (Axon Instruments) and a custom written software (Detector, courtesy J.R. Huguenard, Stanford University), as previously described[94]. Briefly, individual events were detected with a threshold-triggered process from a differentiated copy of the real trace. For each cell, the detection criteria (threshold and duration of trigger for detection) were adjusted to ignore slow membrane fluctuations and electric noise while allowing maximal discrimination of sIPSCs. Detection frames were regularly inspected visually to ensure that the detector was working properly. Comparisons were made among neurons recorded from animals from 1 to 2 litters performed in the same or consecutive days to favor stable recording conditions and minimize variability. We observed no difference in series resistance between groups included in the same statistical analysis. However, we detected a higher series resistance in experiments described in Fig. 3G compared to all other experiments described in Fig. 3, likely explaining the low sIPSCs amplitude in Fig. 3G.

**Fiber photometry**. We used custom-made optical fiber implants of multimode optical fiber (0.39 NA 400 μm core diameter, Thor Labs) inserted in a 1.25 mm diameter 6.4 mm long ceramic ferrule (Thor Labs). In order to target hippocampal CA1, fibers were cut leaving approximately 2 mm beyond the end of the ferrule, polished using polishing sheets of decreasing grit size. Implants were discarded if light transmission was below 70%. Optical implants were positioned with the help of a stereotaxic frame with the tip of the fiber above the CA1. Implants were firmly attached to the skull using light cured glue (Optibond™ Universal, Kerr dental, Bioggio, Switzerland) and dental cements (Unifast™ LC, GC America Inc, Chicago, IL, USA and Kemdent's Simplex Rapid Powder and Liquid, Associated Dental Products Ltd., UK). AAV delivery and fiber implantation took place in the same surgery session. Tip fiber position and AAV infection was verified histologically at the end of the experiment. Before recording sessions, animals were habituated to the recording arena, a 35 × 24 cm plastic enclosure, for 5 days. Experiments were performed in soundproof environment with constant illumination (75 lx). On the recording days, 90 min after the last i.p. injection, animals were connected to a Tucker-Davis Technologies fiber photometry system equipped with 405 and 465 nm LED light sources (Doric Lenses). Photons were collected with a New Focus 2151 photomultiplier and signals digitized using a RZ5P Processor (Tucker-Davis Technologies) and Synapse software. Mouse speed was tracked using ANY-maze software at 15 frames per second to calculate instantaneous acceleration. GCaMP6m processing (detrend and d$F/F$ calculation) via subtraction of the isosbestic control was performed using a manufacturer provided code for MATLAB (MathWorks) that can be found at https://www.tdt.com/docs/sdk/offline-data-analysis/offline-data-matlab/fiber-photometry-epoch-averaging-example/. Traces for immobility/locomotion transitions (Fig. 6D) were calculated using mobility/immobility events tracked by ANY-maze software and aligning the signals of interest (namely calcium-dependent fluorescence and acceleration), using a Python script. The acceleration modulation index (Fig. 6F) was calculated in each experimental mouse using a quadratic fitting of the acceleration–deceleration/d$F/F$ (Fig. 6E) curves in each treatment condition.

**In vivo electrophysiology**. Mice were implanted with fixation head bars under isofluorane. Two silver wires previously chlorinated were inserted over the cerebellum for reference/ground connections. Implant and wires were fixed to skull with light-cured glue (Optibond™ Universal, Kerr dental, Bioggio, Switzerland) and secured with dental cement (Unifast™ LC, GC America Inc., Chicago, IL, USA). Once animals recovered from anesthesia, they were returned to their home cages. Mice were habituated over one week to head-fixation (4 days, 2 sessions per day), where they were allowed to run on top of a wheel (7.5 cm radius) coupled to a stereotactic frame. The day before the first recording, mice were anesthetized and cranial windows were opened at −2 mm posterior from Bregma and ±1.25 mm lateral from midline. Afterwards, craniotomies were covered with low toxicity silicone elastomer (Kwik-Sil™, World Precision Instruments, Sarasota, FL, USA). For recordings, we used 16-channel silicon probes consisting in a linear array with 100 μm resolution and 413 μm² electrode area (Neuronexus). Extracellular signals were pre-amplified (4× gain) and recorded with a 32-channel AC amplifier (100×, Multichannel Systems, Reutlingen, Germany), and sampled at 20 kHz/channel (Digidata 1440, Molecular Devices, San Jose, CA, USA). Analysis of electrophysiological signals was implemented in MATLAB 2019a (MathWorks). For

detection of sharp-wave ripple (SWR), LFP signals from SP were bandpass filtered (70–250 Hz) and smoothed by a Gaussian kernel. Candidate events were detected by thresholding (>3 SDs) and visually validated. The mean ripple power was measured as the mean number of SD of the filtered and smoothed signal from SP during each ripple, compared to the filtered and smoothed signal during non-movement periods. The maximal ripple power considered the peak of the filtered and smoothed signal for each ripple instead of the mean signal.

For theta (4–10 Hz) and gamma (30–90 Hz) oscillations data segments with continuous theta were identified from different recording sessions using spectral criteria. For analysis of the entire frequency band (1–1000 Hz), a Hamming window and the fast Fourier transform (FFT) at 0.5 Hz resolution were used. For analysis of gamma activity, spectral power was estimated from the FFT from 30 to 90 Hz. The contributions of 50 Hz and harmonics were filtered out, and data between the filter limits were interpolated.

**Novel object location test**. Mice were handled and habituated to an open field enclosure (35 × 24 cm) for 5 days before the task. Intraperitoneal treatments also started 5 days before the task. Additionally, mice were habituated to objects by introducing a plastic building block in the home cage. The enclosure was situated in an evenly illuminated, soundproofed box with visual cues in the surrounding walls. The apparatus was cleaned with acetic acid (0.03%) between trials to minimize odor cues. The day of the task, 90 min after the last injection, mice were returned to the open field arena and let to familiarize for 10 min with two identical objects placed at about 5 cm of same long-edge corners. The test session was performed 10 min later by reintroducing mice for 10 min in the open field where one of the objects have been displaced to the corner situated diagonally from the non-displaced object (Fig. 8A). Mouse behavior was recorded and head position tracked using ANY-maze software. Interaction with objects was automatically scored using animal's head entries in the 2 cm perimetral zone surrounding the objects. A discrimination index was calculated as the ratio of the difference in exploration between the displaced and familiar object divided by total exploration.

**Statistical analysis**. All values are given in mean ± SEM, except when noted. Standard $t$ tests were performed to compare Gaussian distributions while Mann–Whitney tests were used for non-Gaussian distributions. One- or two-way ANOVA with repeated-measures followed by Bonferroni's post hoc test or Kruskal–Wallis followed by Dunn post hoc test were used when noted. Group discrimination indices in NOL experiments were tested against chance levels (DI = 0) using one sample $t$ tests. Where appropriate, statistical tests were always two-tailed. For all tests, we adopted an alpha level of 0.05 to assess statistical significance. We provide the exact $p$ value for all the statistical tests except for values below 0.0001 that were indicated by $p < 0.0001$. Statistical analysis was performed using Prism (Graphpad software).

**Reporting summary**. Further information on research design is available in the Nature Research Reporting Summary linked to this article.

## Data availability

The raw data that support the findings of the current study are available from the corresponding author upon reasonable request. Source data are provided with this paper.

## Code availability

GCaMP6m signal processing was performed using a manufacturer provided code for MATLAB (MathWorks) that can be found at https://www.tdt.com/docs/sdk/offline-data-analysis/offline-data-matlab/fiber-photometry-epoch-averaging-example/. The codes used for fiber photometry analysis (alignment to movement start and end, Fig. 6D) and in vivo electrophysiological data analysis (Fig. 7) are available from the corresponding author upon request.

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

## Acknowledgements

We thank E. Jiménez, A. Arroyo, and C. Sanmartín Segovia for help with image analysis; C. Sánchez for Python data processing scripts, J.G. Yagüe and M.A. Arévalo for production and validation of aromatase antibody; A.P. Arnold (UCLA, USA) for the kind gift of the FCG mice and A. Bacci (ICM, Paris, France) and L.M. García-Segura (Cajal Institute, Madrid, Spain) for helpful discussions on the manuscript. This work was supported by grants: RYC-2015-18545 (to P.M.), funded by MCIN/AEI/ 10.13039/501100011033 by "ESF Investing in your future", BFU2017-84490-P (to P.M.) and RTI2018-098581-B-I00 (to L.M.P.) funded by MCIN/AEI/ 10.13039/501100011033 by "ERDF A way of making Europe" and PID2020-112824GB-100 (to P.M.) funded by MCIN/AEI/ 10.13039/501100011033. N.C.-A. is supported by the Ph.D. fellowship PRE2018-084857 funded by MCIN/AEI/10.13039/501100011033 by "ESF Investing in your future". A.S.-A. is supported by the Juan de la Cierva program FJCI-2017-32719 funded by MCIN/AEI/10.13039/501100011033.

## Author contributions

P.M. conceptualized and designed the project, A.H.-V., N.C.-A., A.S.-A., I.A., and P.M. performed and analyzed experiments. A.S.-A. and L.M.P. designed and interpreted silicon probe experiments. A.G.-A. and A.R.-F. performed experiments. P.M. wrote the manuscript with the help for edition and discussion from all authors.

## Competing interests

The authors declare no competing interests.
