## [Peer Review File · Nature Communications]

REVIEWER COMMENTS

Reviewer #1 (Remarks to the Author):

This study studied how the brain-derived Estradiol modulates synaptic inhibition in females' CA1 PV interneurons (PV-IN). Using slice recording of hippocampus pyramidal neurons, calcium recording of PV neurons during locomotion, and sharp-wave ripple recording during the perturbation of brain-derived Estradiol, this study provides potentially exciting insights into the function of hippocampal PV neurons in female mice. Overall, the manuscript will be of broad interest to Nature Communications readers.

However, the findings of PV-IN as mediators of estrogenic regulation of behaviorally relevant female brain activity are much biased (with major concerns described below) and overstated (with below minor concerns). In summary, this study represents an interesting piece to understand hippocampal inhibition by brain sex hormone. The manuscript is suitable for publication in Nature Communications if these issues have been well addressed.

Major concerns:

- 1) The functional role of brain-derived Estradiol in behaviors is not well investigated. For example, any brain neurons might be affected by locomotion, and only the sharp-wave ripple recording is related to the functional role of hippocampus - memory. Without behavioral validation, it is unclear how the manuscript can provide new biological insight.
- 2) Choosing PV cells as the target of studying brain-derived Estradiol in the hippocampus is not well demonstrated, and the specificity of these neurons need to be better documented. CA1 pyramidal and other inhibitory neurons also express aromatase that regulates synaptic plasticity. The authors may need better state why exploring the estrogenic effects on the hippocampal inhibitory PV cells is valuable. For example, how many aromatase mRNAs are expressed in PV cells compared to pyramidal and other inhibitory subtype neurons. The authors should use pyramidal cells and other inhibitory cells as controls.
- 3) The sex-specific role of PV-IN needs to be better validated. In the manuscript, few discussions about the female-biased regulation of hippocampus-related behaviors and no *in vivo* experiments were conducted in the male animals. For example, the authors may need to tell readers to understand if the aromatase or estrogen receptor gene is expressed differently in the CA1 cells (especially the CA1 PV-INs) of male and female mice?

Minor concerns:

- 1) There is no statistic summary of how many Aromatase+ cell-expressed PV and how many PV cells expressed Aromatase+ in Figure 1, and this makes the conclusion of 'estradiol synthesis by this IN population' (lines 108-109) hard to be fully validated. The vague description 'many of these cells were

also positive...' and '...show large variability' in lines 103-104 are inappropriate. Quantification of the mRNA and immunohistochemical expression level in Figure 1 are needed.

2) In lines 136-137, "with a variable range of WFA staining intensities, likely reflecting plasticity of these extracellular structures," literature or evidence may need to be added here to support the conclusion.

3) In Figure 3, the sIPSC amplitude in intact female mice treated with or without β E2 in Fig 3E is far lower than that of mice treated with aromatase blocker letrozole (Arom Block, Fig. 3B, 3C). The authors need to provide some discussion about the results. For the whole-cell patch-clamp recordings experiments (Figures 3-5), sIPSC frequency and amplitude were measured, and only frequency changed (Fig. 3E). The authors may discuss the implication of why no IPSCs amplitude change.

4) In Figure 4, the specificity of ChABC degradation of PNN enwrapping PV-IN rather than other cells needs to be better illustrated.

5) Description in the method part may need to be added why intraperitoneal injection of the aromatase blocker Letrozole for five days rather than a shorter time (like 1-3 days). The long-term IP injection may affect the basic behavioral performance or states of these mice.

Reviewer #2 (Remarks to the Author):

The manuscript by Hernandez-Vivanco et al. described experiments aimed at characterizing the regulation of inhibitory synapses by aromatase in a sex-dependent manner in hippocampal area CA1. By employing genetic, molecular, and electrophysiological tools, the authors found that, similar to CA1 pyramidal neurons (PNs), CA1 parvalbumin-expressing interneurons (PV-INs) express aromatase and regulate synaptic inhibition on PNs through a mechanism that may include perineuronal nets (PNNs) on the PV-INs. Aromatase, likely through its ability to synthesize estradiol, was also found to facilitate the coupling of PV-IN activity with locomotion and modulate the dynamics of sharp wave ripples. The findings expand our current knowledge on the less characterized actions of estradiol (both peripheral and brain-derived) on hippocampal synaptic inhibition and lend support for the idea that inhibitory interneurons and their synaptic connections can be modulated by estradiol, possibly contributing sex-specific, hippocampal-mediated behaviors. In addition, these results broaden our general understanding on how aromatase and aromatase inhibitors, a common therapy in the treatment of cancer, may underlie the cognitive deficits that patients who take these drugs exhibit by altering the inhibitory circuitry in the hippocampus. The experiments appear to be well designed and presented. However a number of problems with the writing weakens the manuscript substantially. Particularly problematic are the overstating of some of the findings (drawing causal inferences where they are unwarranted, for example) and the large number of grammatical errors. Other issues that the authors should consider are listed below.

Major:

- 1) Figure 1: Although the staining for the aromatase protein colocalization looks compelling, the Cyp19a1 mRNA staining in 1A, and to some extent 1B, looks like the red signal could easily be from the Pyramidal cells. Is the colocalization with green PV stain significantly greater than when the (red) image is rotated (one way to distinguish between dots colocalizing with certain cells by chance)?
- 2) A number of results are overstated. For example, the title: the study indeed showed that aromatase regulates CA1 inhibition and network activity in a sex-specific manner, but causality was not demonstrated for de novo estradiol, even though its effects were implied by the presence of aromatase in INs. Same for Lines 27, 200, 370 and 420 : De novo estradiol levels were not measured to show causality, and in only one case was it administered to 'rescue' the effects of aromatase blockade (Figure 3).
- 3) SATB1 staining is very faint and not very informative in Figure 2D, E, F. Furthermore, it is unclear whether this small effect (SATB1+ cells have higher aromatase staining than SATB1- cells) will hold up with a larger number of animals (even though there were a large number of cells analysed, only 3 animals were used). For a discussion of n's, see Lazic, et al. PLoS Biol 2018 Apr 4;16(4):e200528 doi:10.1371/journal.pbio.2005282. eCollection 2018 Apr. "What exactly is 'N' in cell culture and animal experiments? "
- 4) Another example where the results are inaccurately stated is on Line 240, saying that "This result suggests that blockade of estradiol synthesis promotes the plasticity of PV-IN PNN in female mice.". This is speculation that is inappropriate for a results section. In fact, increasing PNNs often leads to decreases synaptic plasticity. Likewise, on line 253, delete the words 'plasticity is'. It is sufficient and more appropriate to just say, 'Our results suggest that PNNs are required for ...'. Additionally, it is unclear how PNNs around IN cell bodies would regulate inhibitory synapses on PCs, so this should be discussed. As an aside, here and throughout, it should be PNNs, not PNN).
- 5) Similarly for Figure 4 and throughout: describe the results, not the interpretation. Plasticity is an interpretation in this case. "PNNs are required for extragonadal aromatase regulation of CA1 synaptic inhibition" is appropriate because the authors disrupted them with chondroitinase, but 'PNN plasticity' is not a good descriptor of manipulation or finding.
- 6) The results plotted in 3B and C should be analysed with a two-way ANOVA for each of the freq data and amplitude data with the variables Ovex/intact and C/Arom block for each.
- 7) Line 177. The authors should stain for estradiol in their Aromatase Block animals and show that estradiol really is down in their hands. Likewise, is 5 days sufficient to get rid of all the estradiol? Again, the authors should provide an image from the timepoint when the physiology was performed.
- 8) Line 184. Again, the authors should show staining for beta E2 from one of each of the treatment groups to estimate (visualize) how much estradiol is really present.
- 9) The methods will need more details. See below.
- 10) The manuscript has grammatical errors throughout. I've noted many of them below, but the authors may need to enlist editorial help.

Minor Issues:

- Summary, second sentence: Based on the context, I think the authors are intending 'Cognitive function relies on ...' (not "relays" on). (also for line 51).
- Line 27: Consider using INs throughout the manuscript where appropriate. Also again, causality was not demonstrated for brain estradiol. It was shown for aromatase the enzyme. (see similar for Lines 29 and 30).
- Similarly, consider using sIPSCs throughout, as they are usually plural.
- Line 41: estradiol is not a feminine hormone per se. It is also produced by males albeit at smaller concentrations.
- Line 51. Consider using an alternative word for 'Correct'. I suggest 'normal'.
- Line 70: 'physiological state' is too general. Which physiological state is being referred to?
- Line 73: "...activities underlying cognitive..." should be 'underlying'
- Line 94. I suggest using the term brain 'sections' because brain 'slices' might be interpreted by the readers as thicker acute brain slices similar to what was used for in vitro electrophysiology.
- Line 137... although suggestive, variability of WFA stain is not evidence of plasticity. Please delete this phrase.
- Line 141: Consider rewording the sentence- it reads like this should add up to 100%, but I think that is not what the authors mean to convey. Please clearly describe the plots in the figures. Same for line 143.
- Line 178: keep capitalization consistent.
- Line 179. I suggest using 'acutely prepared brain slices'.
- Please clarify whether the drug acts on gonadal aromatase or only brain aromatase.
- Figure 3A figure legend: 'Synaptic' is misspelled in 'spontaneous Inhibitory Post-Synaptic Currents (sIPSCs)'
- As mentioned above, please replace "PNN" with 'PNNs' where referring to multiple cells (almost all cases).
- Line 343. "We simultaneously tracked mice position and..." should be 'mouse'. (also on line 700)
- Figure 7: provide rationale for using intact versus OVX animals, especially for experiments in figures 6 and 7.
- Other grammatical errors on lines 225, 236, 399, 476, 489, 491, and 543.

Figures and Legends:

- Line 113: although the authors state that figure 1A represents region CA1, this figure should include a panel that shows where in CA1 this image came from. Also, show PV and aromatase mRNA as individual channels. It is hard to see red dots in PV neurons. Same for Line 115, 1B: Show where in CA1 these neurons were found.
- Line 117: label as 'positive control.'
- Line 120: figure 1E shows presence of estradiol not an accumulation necessarily.
- Line 148: Aromatase labels look cut off on the bottom.
- Figure 2D: please make these green and red to match the legend, or say in the legend that they are shown in greyscale on the left.
- Ovex control sIPSC freq is almost double intact sIPSC freq. The 2 frequency graphs and 2 amplitude graphs are on different Y axes. All should be plotted and analyzed together, as it will almost certainly reveal a significant main effect of Ovex.
- Figure 3A: what does the '14d' label represent? Did treatment begin 14 days after ovariectomy?
- Really, more than 3 animals should be used for this data. This might be a good opportunity to provide images as well for this treatment.
- Line 264: explain the '14d' label in figure 4C.
- Figure 5. please stain for estradiol here too. Is that staining decreased in Arom-blocked males?

Methods:

- It is somewhat surprising that estrous cycle was neither tracked (nor reported), especially given the claim that peripheral and de novo estradiol might have cooperative actions on the effects observed (experiments for figure 3). There are reports that de novo production of sex steroid levels in the hippocampus show cyclicity as gonadal sex steroids. It would be important to know the systemic hormonal milieu that intact animals were in (high estradiol estrus stages vs. non) for the experiments described in this study and how this might contribute to the effects observed. Otherwise, the authors will need to provide a statement as to why they thought it was unnecessary and discuss the implications.
- Line 588: provide age range and number of animals for each of the experiments described. This information is not consistently reported throughout the manuscript.
- Line 610: when did treatment begin post-surgery?
- Line 638: report number of sections/animal/group that were quantified. Were section averaged/animal?
- Line 647. "After 3 more step of washing..." should be 'steps'.

- Line 660: how long after last treatment were slices prepared? Report number of animals per group.
- Line 670. “GABA receptors mediated inhibitory spontaneous (IPSCs)...” should be ‘GABA receptor-mediated spontaneous Inhibitory PostSynaptic Currents (sIPSCs)...’
- Please clearly specify in Methods that recordings were made from pyramidal neurons.
- Methods need to contain more details for reproducibility and replicability purposes. Were experimenters doing electrophysiology or image analyses blinded to treatment, for example? Please state either way.

Reviewer #3 (Remarks to the Author):

The current report is a comprehensive, carefully conducted and important characterization of the roles of ovarian and extragonadal estrogens in regulation of the activity parvalbumin-expressing inhibitory neurons (PV-IN) in CA1 of the hippocampus. Results reveal expression of aromatase mRNA and protein in PV-IN basket cells, to my knowledge, for the first time. Such expression suggest a mechanism by which locally synthesized estrogens impact hippocampal inhibitory activity. Results demonstrate that estradiol acts to regulate synaptic inhibition in CA1 pyramidal cells by suppressing function of perineuronal nets (PNN) in the PV-IN. Furthermore, the functional significance of this regulation was revealed in vivo. Blocking estradiol synthesis via systemic administration of an aromatase inhibitor increases coupling of PV-IN activity to locomotion and hippocampal network activity influences by PV-IN (i.e. sharp-wave ripples). Importantly, they demonstrate this regulation occurs in females, but not male mice and through the use of the Four Core Genotype mice, demonstrate the sex difference is regulated by gonadal hormones and not genetic sex.

Overall, the data appear to be technically sound and appropriately reported. There is a growing interest in understanding the role of locally synthesized estrogens in the regulation of hippocampal function. These comprehensive and convincing results, providing evidence of extragonadal estrogen regulation of PV-IN in the hippocampus, are very interesting, significant, and have the potential to move the field forward. I have the following suggestions for improvement.

1. A major concern that I have with the study is that the authors use systemic administration of letrozole (aromatase inhibitor). Though ovariectomy eliminates the major source of extragonadal circulating estrogens, it does eliminate all sources. Therefore, the conclusions made about hippocampal “local estradiol” throughout the paper are overstated, in my view. Only with the use of brain specific administration of letrozole can that claim be made. At least one demonstration that icv administration of letrozole provides similar effects as systemic administration would strengthen the conclusions.

2. I find the result that systemic exogenous estradiol administration in the presence of letrozole rescues effects of aromatase blockage in OVX mice surprising (e.g. Fig 3D) and suggestive of no difference in the roles of circulating and locally synthesized estrogens in these effects. That is somewhat in contradiction to current views of the relationship between these sources of the hormone. There is some evidence that effects mediated by neuroestrogens vs ovarian estrogens occur due to the higher levels of the hormone produced locally as compared to peripheral sources and /or rapid local release. Additionally, circulating estradiol may regulate production of brain estradiol through its regulation of brain aromatase. There is much interest in understanding the respective roles of these sources of estradiol. It is unclear how the current results fit in with the current understanding of these roles. It would be helpful to understand what the authors believe is the relationship between circulating and brain-derived estradiol and how do their results add to that knowledge.

3. In a number of places in the manuscript, the authors state the lack of previous results regarding the role of estradiol in the regulation of inhibitory activity in the hippocampus. However, major work coming out of Catherine Woolley's lab has addressed this issue and should be discussed. Although the work is cited (Ref 21 and 23), the relevant results are not discussed. Of particular importance is previous work that identified a sex difference in estradiol regulation of inhibition in the hippocampus with results suggestive of a role for locally synthesized estradiol (21).

In the revised manuscript, changes in text and figures have been highlighted in yellow.

Reviewer #1

We thank the reviewer for considering the manuscript. We are pleased to read that he/she finds it of broad interest and suitable for publication.

Major concerns:

1) *The functional role of brain-derived Estradiol in behaviors is not well investigated. For example, any brain neurons might be affected by locomotion, and only the sharp-wave ripple recording is related to the functional role of hippocampus - memory. Without behavioral validation, it is unclear how the manuscript can provide new biological insight.*

We have performed a new experiment to extend our analysis of sharp-wave ripples (SWRs) to other forms of hippocampal network activity with a prominent behavioral role: gamma and theta oscillations. The results show that aromatase blockade decreases the power of these two forms of hippocampal network activity. Importantly, the effect of aromatase blockade is reversed by simultaneous administration of β E2 (Fig. 7D,E).

In addition, we have performed a new experiment to evaluate the effect of aromatase blockade in the hippocampal-dependent novel object location (NOL) task in OVX female mice and male mice. The results show that aromatase blockade impaired the performance of OVX female mice in the NOL task, an effect that was reversed by simultaneous administration of β E2 (Fig. 8A,B). On the other hand, male mice performance was unaffected by aromatase blockade (Fig. 8C).

These results are included in the current version of the manuscript: Fig. 7 and 8, Results section (lines 455 and 522), Discussion section (lines 687 and 642) and Methods section (lines 907 and 914).

2) *Choosing PV cells as the target of studying brain-derived Estradiol in the hippocampus is not well demonstrated, and the specificity of these neurons need to be better documented. CA1 pyramidal and other inhibitory neurons also express aromatase that regulates synaptic plasticity. The authors may need better state why exploring the estrogenic effects on the hippocampal inhibitory PV cells is valuable. For example, how many aromatase mRNAs are expressed in PV cells compared to pyramidal and other inhibitory subtype neurons. The authors should use pyramidal cells and other inhibitory cells as controls.*

We have performed new analysis of previous experiments to assess aromatase protein expression in excitatory neurons and compare with PV INs. In addition, we have performed a new experiment to assess aromatase expression in a different type of CA1 IN defined by the expression of the marker somatostatin (SST). Although we observed aromatase expression in all three cell types, the expression of aromatase protein was higher in PV INs compared excitatory neurons (Fig. 1F) and SST INs (Fig. S1A,B) in CA1 of female mice.

The current study is focused on PV INs because this IN subtype shows prominent aromatase expression.

These results are included in the current version of the manuscript: Fig 1F, Fig. S1A,B, Results section (line 124) and Methods section (lines 804).

3) *The sex-specific role of PV-IN needs to be better validated. In the manuscript, few discussions about the female-biased regulation of hippocampus-related behaviors and no in vivo experiments were conducted in the male animals. For example, the authors may need to tell readers to understand if the aromatase or estrogen receptor gene is expressed differently in the CA1 cells (especially the CA1 PV-INs) of male and female mice?*

We have performed new analysis of previous experiments and quantified Aromatase protein expression in CA1 PV INs from male and female mice. No significant difference was observed (Fig. S4A). The reviewer makes a very interesting suggestion regarding the expression of estrogen receptors. We are not aware of reliable commercial antibodies against the three different ER subtypes. Experiments using in situ mRNA detection are currently planned to address this relevant aspect of estrogen signaling in the hippocampus.

In addition, we have performed a new experiment to assess the effect of aromatase blockade in hippocampal dependent memory in male and OVX female mice. This *in vivo* experiment reveals that aromatase blockade affects performance on the Novel Object Location test in OVX females

and not in intact male mice (Fig. 8B,C). These results strengthen the conclusion of female specific role of aromatase regulation of CA1 PV INs.

The results of these analysis and experiments are included in the current version of the manuscript: Fig S4, Fig 8, Results section (lines 341 and 532).

We have also extended the discussion on sex-specific effects of aromatase in hippocampal inhibition (line 652).

Minor concerns:

1) *There is no statistic summary of how many Aromatase+ cell-expressed PV and how many PV cells expressed Aromatase+ in Figure 1, and this makes the conclusion of 'estradiol synthesis by this IN population' (lines 108-109) hard to be fully validated. The vague description 'many of these cells were also positive...' and '...show large variability' in lines 103-104 are inappropriate. Quantification of the mRNA and immunohistochemical expression level in Figure 1 are needed.*

We have performed new analysis to quantify aromatase mRNA expression levels in CA1 PV INs, and compared it to CA1 *stratum pyramidale* and medial amygdala neurons (MeA). We found lower levels of aromatase mRNA in PV CA1 INs compared with MeA neurons but higher compared with *stratum pyramidale* (Fig. 1D).

In addition, we have performed new analysis that quantitatively describes aromatase protein expression in PV+ INs and excitatory neurons cell bodies from the CA1 Pyramidal (PYR) cell layer. This analysis shows that expression of aromatase protein is higher in PV INs compared with excitatory neurons (Fig. 1F).

The results of these analysis are included in the current version of the manuscript: Fig 1D, Fig. 1F, Results section (lines 120 and 124).

2) *In lines 136-137, "with a variable range of WFA staining intensities, likely reflecting plasticity of these extracellular structures," literature or evidence may need to be added here to support the conclusion.*

In accordance with this comment and a minor comment from reviewer #2, we have deleted the last part of the sentence. We agree with both reviewers that variability does not reflect plasticity. Please, see reviewer #2 major comments 4 & 5 for the use of the term "PNN plasticity" in the revised version of the manuscript.

We have deleted the last part of the sentence and use alternative terms to describe the changes observed in PNNs. (Lines 44, 289 and 675).

3) *In Figure 3, the sIPSC amplitude in intact female mice treated with or without β E2 in Fig 3E is far lower than that of mice treated with aromatase blocker letrozole (Arom Block, Fig. 3B, 3C). The authors need to provide some discussion about the results. For the whole-cell patch-clamp recordings experiments (Figures 3-5), sIPSC frequency and amplitude were measured, and only frequency changed (Fig. 3E). The authors may discuss the implication of why no IPSCs amplitude change.*

The lower sIPSC amplitude in β E2 treated intact female mice (Fig. 3H in the revised manuscript) compared experiments in intact and OVX female mice (Fig. 3D, F in the revised manuscript) is likely due to difference in recording conditions between the two sets of experiments, in particular to a higher series resistance in experiments in Fig. 3H (see figure below).

In the manuscript, all comparisons were made among neurons recorded from animals from 1 or 2 litters performed in the same or consecutive days to favor stable recording conditions and minimize variability. Series resistance was constantly monitored in all experiments. We observed no difference in series resistance between groups included in the same statistical analysis.

This information is now included in the Methods section (line 852).

In addition, we have discussed the implications of the reported changes in sIPSC frequency but not in amplitude in response to aromatase blockade (Discussion section, lines 622 and 627).

4) In Figure 4, the specificity of ChABC degradation of PNN enwrapping PV-IN rather than other cells needs to be better illustrated.

We agree that ChABC may degrade PNN irrespectively of cell type. However, in our analysis, we found no CA1 PV- neurons enwrapped by PNNs. Instead, our analysis suggests that in CA1, virtually all WFA+ cells are PV+ INs. These data are in agreement with previous studies of PNN distribution in dorsal CA1 (Yamada and Jinno, JCN, 2014).

We have included this information in the manuscript (Results section, line 175).

5) Description in the method part may need to be added why intraperitoneal injection of the aromatase blocker Letrozole for five days rather than a shorter time (like 1-3 days). The long-term IP injection may affect the basic behavioral performance or states of these mice.

Prolonged aromatase inhibition (5-7 days) is required to maximally block hippocampal synaptic plasticity (i.e. LTP) in intact and OVX female mice (Vierk et al, *J Neurosci*, 2012; Wang et al, *J Neurosci*, 2018).

We have included the rationale for prolonged aromatase inhibition in the Results section (line 217).

Reviewer #2

We thank the reviewer for his/her thoughtful comments on our work. We were pleased by his/her positive comments about the design and presentation of the experiments.

Major:

1) Figure 1: Although the staining for the aromatase protein colocalization looks compelling, the Cyp19a1 mRNA staining in 1A, and to some extent 1B, looks like the red signal could easily be from the Pyramidal cells. Is the colocalization with green PV stain significantly greater than when the (red) image is rotated (one way to distinguish between dots colocalizing with certain cells by chance)?

We have performed new analysis to quantify aromatase RNA expression in CA1 PV IN. As per reviewer's suggestion, we have included an additional control to evaluate chance levels of colocalization in CA1 PV INs. Signals from aromatase mRNA and PV-INs showed significantly higher colocalization levels compared with chance colocalization observed after relative rotation of the images.

This analysis is included in the revised version of the manuscript, Results section (line 114) and Methods section (line 817).

2) A number of results are overstated. For example, the title: the study indeed showed that aromatase regulates CA1 inhibition and network activity in a sex-specific manner, but causality was not demonstrated for de novo estradiol, even though its effects were implied by the presence of aromatase in INs. Same for Lines 27, 200, 370 and 420 : De novo estradiol levels were not measured to show causality, and in only one case was it administered to 'rescue' the effects of aromatase blockade (Figure 3).

We completely agree with the reviewer in the importance of experiments administering β E2 (recovery experiments) in providing evidences of the involvement of β E2. For this reason, in addition to the two recovery experiments included in the initial version of the manuscript (slice

electrophysiology Fig. 3 and *in vivo* fiber photometry Fig. 6), we have performed three new recovery experiments. The revised manuscript now includes:

- 1) The recovery condition in the *in vivo* Sharp Wave Ripples (SWRs) electrophysiology experiments (Fig. 7). The analysis of this new experiment shows that the effect of aromatase blockade on SWR rate and power are reverted to control levels by simultaneous treatment with β E2 (Fig. 7D).
- 2) We have performed new *in vivo* electrophysiological recordings and analysis of two other prominent forms of hippocampal network activity, the theta and the gamma oscillations. We now describe the effect of aromatase blockade in decreasing the power of these two forms of hippocampal network activity and show that the effects are reverted by administering β E2 (Recovery, Fig. 7E).
- 3) We have used the Novel Object Location (NOL) test to determine the cognitive effects of brain aromatase blockade and β E2 treatment on hippocampal memory. This new experiment shows that aromatase blockade reduces the performance of OVX female mice in NOL test to chance levels. β E2 simultaneous treatment (recovery) resulted in performance over chance levels, similar to untreated mice (Fig. 8B).

Thus, recovery with β E2 was observed in aromatase blockade effects *ex vivo* (sIPSC analysis) and *in vivo* (optical monitoring of PV INs activity, SWR, theta and gamma oscillations and hippocampal-dependent memory). Altogether, recovery experiments strongly suggest that β E2 produced by aromatase regulates CA1 synaptic inhibition and support the main conclusion of the study.

The new experiments and analysis are included in the revised version of the manuscript: Fig. 7D-E, Fig. 8B, Results section (lines 455 and 522) and Methods section (lines 907 and 914).

We have modified the manuscript at the indicated lines to accurately reflect aromatase regulation of CA1 synaptic inhibition and PV INs (lines 2, 41, 43, 45, 46, 245, 429, 491 and 1246).

3) SATB1 staining is very faint and not very informative in Figure 2D, E, F. Furthermore, it is unclear whether this small effect (SATB1+ cells have higher aromatase staining than SATB1- cells) will hold up with a larger number of animals (even though there were a large number of cells analysed, only 3 animals were used). For a discussion of n's, see Lazic, et al. PLoS Biol 2018 Apr 4;16(4):e200528 doi:10.1371/journal.pbio.2005282. eCollection 2018 Apr. "What exactly is 'N' in cell culture and animal experiments? "

Consistent with previous reports (Viney et al, Nat Neurosci 2016; Yamada and Jinno, J Comp Neurol, 2015), we observed nuclear and sparse SATB1 staining in CA1, as reflected in the image in Fig. 2D. The quantification of aromatase staining in SATB1+/- PV-INs shows a statistically highly significant effect (Fig. 2E). Moreover, using a similar threshold based approach applied to Fig. 2 C, F, we found a higher proportion of aromatase+ in PNN+ and SATB1+ PV-Ns compared with PNN- and SATB1- PV-INs using individual mice as n (see figure below). For these reasons, we believe that the results of the analysis of aromatase expression in SATB1+ PV- INs support our conclusion of higher aromatase expression in PNN+ SATB1+ PV basket INs.

We did not modified the manuscript in response to this comment.

Left graph. Proportion of aromatase expressing WFA+ and WFA- PV INs. Two-way ANOVA $F(1, 8) = 91.88, p < 0.0001$. Bonferroni's multiple comparisons test, $p = 0.003$. $n = 3$ female mice.

Right graph. Proportion of aromatase expressing SATB1+ and SATB1- PV INs. Two-way ANOVA $F(1, 8) = 19.3, p = 0.023$. Bonferroni's multiple comparisons test, $p = 0.029$. $n = 3$ female mice.

4) Another example where the results are inaccurately stated is on Line 240, saying that “This result suggests that blockade of estradiol synthesis promotes the plasticity of PV-IN PNN in female mice.”. This is speculation that is inappropriate for a results section. In fact, increasing PNNs often leads to decreases synaptic plasticity. Likewise, on line 253, delete the words ‘plasticity is’. It is sufficient and more appropriate to just say, ‘Our results suggest that PNNs are required for ...’. Additionally, it is unclear how PNNs around IN cell bodies would regulate inhibitory synapses on PCs, so this should be discussed. As an aside, here and throughout, it should be PNNs, not PNN). We agree with the reviewer, the term plasticity was inappropriately used to describe changes in PNNs observed after aromatase blockade.

We have edited the manuscript at the indicated lines to better describe the results of the experiments (Results section, lines 296 and 309).

In addition, we discuss how PNNs around PV INs may be regulating inhibitory synapses on pyramidal neurons (Discussion section, line 623) and use PNNs instead of PNN when appropriate in the revised manuscript (several instances, highlighted).

5) Similarly for Figure 4 and throughout: describe the results, not the interpretation. Plasticity is an interpretation in this case. “PNNs are required for extragonadal aromatase regulation of CA1 synaptic inhibition” is appropriate because the authors disrupted them with chondroitinase, but ‘PNN plasticity’ is not a good descriptor of manipulation or finding.

We have edited the manuscript as per reviewer's suggestion. (Results section, line 309).

6) The results plotted in 3B and C should be analysed with a two-way ANOVA for each of the freq data and amplitude data with the variables Ovex/intact and C/Arom block for each.

After verifying that recording conditions were comparable between intact and OVX recordings (see Reviewer #1, minor comment #3), we have performed new statistical analysis suggested by the reviewer. This analysis shows a significant effect of both variables (OVX/Intact and Control/Arom Block) for sIPSC frequency but not sIPSC amplitude.

The analysis is reported in Fig. 2D (line 255).

7) Line 177. The authors should stain for estradiol in their Aromatase Block animals and show that estradiol really is down in their hands. Likewise, is 5 days sufficient to get rid of all the estradiol? Again, the authors should provide an image from the timepoint when the physiology was performed.

Although tempting at first, experiments aimed at providing a quantitative measure of estradiol levels using immunohistochemistry may not yield reliable results. Likely due to the small and lipophilic nature of estradiol and the impact of fixation and detergent treatment, the variability of estradiol staining hinders the use of the procedure for quantitative purposes. In addition, this procedure does not address the well documented and potentially important role of mechanisms regulating estradiol release in the regulation of CA1 synaptic inhibition. However, we agree with the reviewer on the importance of addressing the role of β E2. With this purpose, we have reinforced the recovery approach in the *ex vivo* and *in vivo* approaches (see response to major comment #2).

The recovery experiments are now shown in Fig. 3G, 6E,F, 7D,E and 8B.

8) Line 184. Again, the authors should show staining for beta E2 from one of each of the treatment groups to estimate (visualize) how much estradiol is really present.

Recovery experiments were performed in OVX to address the involvement of β E2 in mice lacking the main peripheral estrogen producing organ.

The recovery experiments are now shown in Fig. 3G, 6E,F, 7D,E and 8B.

9) The methods will need more details. See below.

We have added information to the methods section as per reviewer's suggestions (see specific comments).

10) The manuscript has grammatical errors throughout. I've noted many of them below, but the authors may need to enlist editorial help.

We have corrected the grammatical errors, for which we apologize. We thank the reviewer for his/her efforts in improving this aspect of the manuscript.

Minor Issues:

- Summary, second sentence: Based on the context, I think the authors are intending 'Cognitive function relies on ...' (not "relays" on). (also for line 51). **We have edited the manuscript as per reviewer's suggestion (lines 39 and 65).**

- Line 27: Consider using INs throughout the manuscript where appropriate. Also again, causality was not demonstrated for brain estradiol. It was shown for aromatase the enzyme. (see similar for Lines 29 and 30). **We have used INs where appropriate in the manuscript (several instances, highlighted).**

The results of the recovery approach reported in Figs. 3G, 6E,F, 7D,E and 8 strongly suggest that β E2 produced by aromatase regulates CA1 synaptic inhibition and the function of PV INs. We have modified the manuscript to accurately reflect aromatase regulation of CA1 synaptic inhibition and PV INs (lines 2, 41, 43, 45, 46, 245, 429, 491 and 1246).

- Similarly, consider using sIPSCs throughout, as they are usually plural. **We have edited the manuscript as per reviewer's suggestion (several instances, highlighted).**

- Line 41: estradiol is not a feminine hormone per se. It is also produced by males albeit at smaller concentrations. **We have clarified that estradiol is a sex hormone involved in the control of female reproductive system (line 54).**

- Line 51. Consider using an alternative word for 'Correct'. I suggest 'normal'. **We have edited the manuscript as per reviewer's suggestion (line 65).**

- Line 70: 'physiological state' is too general. Which physiological state is being referred to? **We have defined the physiological states that affect activity of PV INs and added relevant references (line 84).**

- Line 73: "...activities underlying cognitive..." should be 'underlying'. **We have edited the manuscript as per reviewer's suggestion (line 88).**

- Line 94. I suggest using the term brain 'sections' because brain 'slices' might be interpreted by the readers as thicker acute brain slices similar to what was used for in vitro electrophysiology. **We have edited the manuscript as per reviewer's suggestion (line 110).**

- Line 137... although suggestive, variability of WFA stain is not evidence of plasticity. Please delete this phrase. **We have deleted the last part of the sentence (line 174).**

- Line 141: Consider rewording the sentence- it reads like this should add up to 100%, but I think that is not what the authors mean to convey. Please clearly describe the plots in the figures. Same for line 143. **We have rearranged plots in Fig. 2F and made an accurate description of the plots in the Results section (line 177 and 183).**

- Line 178: keep capitalization consistent. **We have edited the manuscript as per reviewer's suggestion (line 220).**

- Line 179. I suggest using 'acutely prepared brain slices'. **We have edited the manuscript as per reviewer's suggestion (line 221).**

- Please clarify whether the drug acts on gonadal aromatase or only brain aromatase. **This information is now included in the revised manuscript (line 216).**

- Figure 3A figure legend: ‘Synaptic’ is misspelled in ‘spontaneous Inhibitory Post-Synaptic Currents (sIPSCs)’. **The typo has been corrected (line 249).**
- As mentioned above, please replace “PNN” with ‘PNNs’ where referring to multiple cells (almost all cases). **We have edited the manuscript as per reviewer's suggestion (several instances, highlighted in the revised manuscript).**
- Line 343. “We simultaneously tracked mice position and...” should be ‘mouse’. (also on line 700). **We have edited the manuscript as per reviewer's suggestion (line 402 and 875).**
- Figure 7: provide rationale for using intact versus OVX animals, especially for experiments in figures 6 and 7. Intact (Fig. 6) and OVX (Fig. 7) animals were used for in vivo experiments for two reasons: 1) similar to the *ex vivo* experiments in Fig. 3D, we aimed to test the in vivo effect of aromatase blockade on CA1 inhibition in intact and OVX female mice, 2) we determined the impact of the recovery treatment with β E2 in the *in vivo* effects of aromatase blockade in intact and OVX female mice. **The rationale is now explained in the manuscript (lines 397 and 464).**
- Other grammatical errors on lines 225, 236, 399, 476, 489, 491, and 543. **The noted and other errors have been corrected. We thank again the reviewer for his/her help in improving this aspect of the manuscript.**

Figures and Legends:

- Line 113: although the authors state that figure 1A represents region CA1, this figure should include a panel that shows where in CA1 this image came from. Also, show PV and aromatase mRNA as individual channels. It is hard to see red dots in PV neurons. Same for Line 115, 1B: Show where in CA1 these neurons were found. **We have clearly indicated the region where images and histological analysis was performed as the dorsal CA1 (line 139, 145, 369, 788, 1191). We have included a panel to show the approximate location (Fig. 1A).**
- Line 117: label as ‘positive control.’ **The figure legend was modified as per reviewer's suggestion (Fig. 1C, lines 119 and 147).**
- Line 120: figure 1E shows presence of estradiol not an accumulation necessarily. **The figure legend and result section was edited as per reviewer's suggestion (lines 129, 138 and 159).**
- Line 148: Aromatase labels look cut off on the bottom. **The figure was modified to avoid cut off (line 189).**
- Figure 2D: please make these green and red to match the legend, or say in the legend that they are shown in greyscale on the left. **The figure legends were edited to describe greyscale images in panel 1A (line 141), 1E (line 154) 1G (line y), 5A (line 160), S1A (line 1191) and S5A (line 1248).**
- Ovex control sIPSC freq is almost double intact sIPSC freq. The 2 frequency graphs and 2 amplitude graphs are on different Y axes. All should be plotted and analyzed together, as it will almost certainly reveal a significant main effect of Ovex. **We have plotted data from intact and OVX female mice in a single graph (Fig. 3D) and analyzed the results as the reviewer suggested in his/her major comment #6 (line 255).**
- Figure 3A: what does the ‘14d’ label represent? Did treatment begin 14 days after ovariectomy? **Treatment begun 14 days after ovariectomy. We have modified figures and the legends to clarify the experimental procedure (Fig. 1A, line 247; Fig. 4C, line 321, Fig. 7A, line 494, Fig. 8B, 546).**
- Really, more than 3 animals should be used for this data. This might be a good opportunity to provide images as well for this treatment. **The finding reported in this figure was replicated in independent experiments, both in intact (Fig. 5F) and OVX female mice (Fig. 3E and 4C). We**

believe that replicating the finding in independent experiments adds consistency to the conclusions making the proposed increase in the *n* less necessary. Regarding the images, we refer to our answer to major comment #7.

• Line 264: explain the '14d' label in figure 4C. **The figure legend was modified to clarify the label in the experimental design schema (line 321).**

• Figure 5. please stain for estradiol here too. Is that staining decreased in Arom-blocked males? **As explained in response to major comment #7, we are not confident on the use of immunodetection of β E2 with quantitative purposes.**

Methods:

• It is somewhat surprising that estrous cycle was neither tracked (nor reported), especially given the claim that peripheral and de novo estradiol might have cooperative actions on the effects observed (experiments for figure 3). There are reports that de novo production of sex steroid levels in the hippocampus show cyclicity as gonadal sex steroids. It would be important to know the systemic hormonal milieu that intact animals were in (high estradiol estrus stages vs. non) for the experiments described in this study and how this might contribute to the effects observed. Otherwise, the authors will need to provide a statement as to why they thought it was unnecessary and discuss the implications.

The reviewer raises a very interesting point. We believe that the experiments in OVX female mice rule out the requirement of peripheral hormones cyclicity in brain aromatase control of CA1 synaptic inhibition. Indeed, experiments in OVX female mice (Fig. 3C,D, Fig. 4C,D, Fig. 7, Fig. 8 and Fig. S3) demonstrate that brain aromatase regulates CA1 synaptic inhibition and PV INs independently of estrous cycle regulation of brain estrogen synthesis. On the other hand, recovery experiments (Fig. 3G, Fig 6F, Fig. 7D, E and Fig. 8) indicate that brain aromatase activity is not required for exogenous β E2 to regulate CA1 synaptic inhibition and hippocampal function. This is in accordance with the well documented independent regulation of brain and ovarian β E2 synthesis. We have discussed the interaction between peripheral and central estrogen sources in the revised manuscript.

The estrus cycle in intact female mice was not monitored. This information has been included in the Method section (line 748). In addition, we have extended our discussion on the independent role of peripheral and brain estradiol in the regulation of CA1 synaptic inhibition (lines 596 and 605).

• Line 588: provide age range and number of animals for each of the experiments described. This information is not consistently reported throughout the manuscript. **We have included information about the age of animals used in the study in the Methods section (line 742). We also report the exact number of animals used in each experiment in the correspondent figure legends.**

• Line 610: when did treatment begin post-surgery? **We have completed the information about the post-surgery delay in the different procedures: AAV delivery, ChABC injection, cannula implantation and ovariectomy (lines 774 and 782 and correspondent figure legends).**

• Line 638: report number of sections/animal/group that were quantified. Were section averaged/animal? **We provide the minimal number of section analyzed per animal and the procedure to average the results for each animal (line 822).**

• Line 647. "After 3 more step of washing..." should be 'steps'. **We have edited the manuscript as per reviewer's suggestion (line 809).**

• Line 660: how long after last treatment were slices prepared? Report number of animals per group. **We have added information about the delay (90 minutes) between the last treatment and slice preparation (line 829). The number of animals is reported for each experiment in the correspondent figure legend.**

• Line 670. “GABA receptors mediated inhibitory spontaneous (IPSCs)...” should be ‘GABA receptor-mediated spontaneous Inhibitory PostSynaptic Currents (sIPSCs)...’ **We have edited the manuscript as per reviewer's suggestion (line 839).**

• Please clearly specify in Methods that recordings were made from pyramidal neurons. **We have clearly indicated that recordings were made from pyramidal neurons (line 834).**

• Methods need to contain more details for reproducibility and replicability purposes. Were experimenters doing electrophysiology or image analyses blinded to treatment, for example? Please state either way. **We have updated the information about study design and analysis of images, in vivo and in vitro electrophysiological data, fiber photometry and behavior in the Reporting Summary file.**

Reviewer #3

We are pleased to read that the reviewer finds the data interesting and significant. We thank the reviewer for his/her suggestions for the improvement of the manuscript.

1. A major concern that I have with the study is that the authors use systemic administration of letrozole (aromatase inhibitor). Though ovariectomy eliminates the major source of extragonadal circulating estrogens, it does (not) eliminate all sources. Therefore, the conclusions made about hippocampal “local estradiol” throughout the paper are overstated, in my view. Only with the use of brain specific administration of letrozole can that claim be made. At least one demonstration that icv administration of letrozole provides similar effects as systemic administration would strengthen the conclusions.

We have performed a new experiment to evaluate the effect of direct brain administration of the aromatase blocker letrozole in CA1 synaptic inhibition and PV INs of OVX female mice. Electrophysiological recording showed that Intracerebroventricular (icv) injections of the aromatase blocker letrozole resulted in increased sIPSC frequency in CA1 pyramidal neurons with respect to vehicle injected mice. Consistent with systemic application in intact and OVX mice, the amplitude of sIPSC was unaffected (Fig. 3E, F).

In addition, we have determined the effect of icv administration of the aromatase blocker in the perineuronal nets surrounding CA1 PV+ INs. We observed that, similar to systemic application, intracerebroventricular letrozole increased PNNs around PV INs of OVX female mice (Fig. S3B). These results strengthen the conclusion on the critical role of brain aromatase in the regulation of CA1 synaptic inhibition and PV+ INs.

These new results are reported in the new version of the manuscript (Fig. 3E,F and Fig. S3B), in the Results (lines 227 and 293), Discussion (line 593) and Methods (line 776) sections.

2. I find the result that systemic exogenous estradiol administration in the presence of letrozole rescues effects of aromatase blockage in OVX mice surprising (e.g. Fig 3D) and suggestive of no difference in the roles of circulating and locally synthesized estrogens in these effects. That is somewhat in contradiction to current views of the relationship between these sources of the hormone. There is some evidence that effects mediated by neuroestrogens vs ovarian estrogens occur due to the higher levels of the hormone produced locally as compared to peripheral sources and /or rapid local release. Additionally, circulating estradiol may regulate production of brain estradiol through its regulation of brain aromatase. There is much interest in understanding the respective roles of these sources of estradiol. It is unclear how the current results fit in with the current understanding of these roles. It would be helpful to understand what the authors believe is the relationship between circulating and brain-derived estradiol and how do their results add to that knowledge.

The main conclusion of the current experiments regarding this very interesting point is that ovarian and brain estradiol (β E2) regulation of CA1 synaptic inhibition and PV INs can occur independently. The evidences that support this conclusion are: 1) Brain aromatase regulation of CA1 synaptic inhibition and PV INs (for example in icv experiments in OVX female, Fig. 3F, Fig. S3B) occurs regardless the peripheral β E2 regulation of brain β E2 production. 2) The new recovery experiments included in the manuscript (sIPSC analysis, Fig. 3G; optical monitoring of PV INs activity (Fig 6F); SWR (Fig. 7D); theta and gamma oscillations (Fig. 7E) and behavior (Fig. 8) indicate that brain aromatase activity is not required for exogenous β E2 to regulate CA1

synaptic inhibition and hippocampal function. This is in accordance with the well documented independent regulation of brain and ovarian β E2 synthesis.

We have reinforced the recovery experiments (Fig. 3G, 6E,F, 7D,E and 8B) and expanded the discussion differential role of peripheral and local estradiol sources according to our experiments (lines 596 and 605).

3. In a number of places in the manuscript, the authors state the lack of previous results regarding the role of estradiol in the regulation of inhibitory activity in the hippocampus. However, major work coming out of Catherine Woolley's lab has addressed this issue and should be discussed. Although the work is cited (Ref 21 and 23), the relevant results are not discussed. Of particular importance is previous work that identified a sex difference in estradiol regulation of inhibition in the hippocampus with results suggestive of a role for locally synthesized estradiol (21).

We completely agree with the reviewer on the high relevance of the findings of Catherine Woolley's laboratory. **We have expanded the discussion of the mentioned references (line 677).**

REVIEWERS' COMMENTS

Reviewer #1 (Remarks to the Author):

The authors have addressed my concerns and the paper is now suitable for publication

Reviewer #2 (Remarks to the Author):

The authors have adequately addressed most of my concerns. I have only one minor issue remaining:

Although 'Estrus cycle' has certainly been used in the literature, 'Estrous cycle' is more commonly used, with 'Estrus' referring to the one stage of the cycle. Either way, the authors need to be consistent in their usage, as both are used here.

Reviewer #3 (Remarks to the Author):

The authors have been responsive to my comments and have adequately addressed my concerns.

REVIEWERS' COMMENTS

Reviewer #1

The authors have addressed my concerns and the paper is now suitable for publication

We thank the reviewer for her/his contribution to improve the manuscript and her/his positive assessment of the work.

Reviewer #2:

The authors have adequately addressed most of my concerns. I have only one minor issue remaining:

Although 'Estrus cycle' has certainly been used in the literature, 'Estrous cycle' is more commonly used, with 'Estrus' referring to the one stage of the cycle. Either way, the authors need to be consistent in their usage, as both are used here.

We thank again the reviewer for her/his help in improving this and other aspects of the manuscript and for the positive assessment of our work.

We have consistently used the term “estrous cycle” throughout the manuscript.

Reviewer #3:

The authors have been responsive to my comments and have adequately addressed my concerns.

We thank the reviewer for her/his contribution to improve the manuscript and her/his positive assessment of the work.